# Phytochemistry, Pharmacology and Mode of Action of the Anti-Bacterial *Artemisia* Plants

**DOI:** 10.3390/bioengineering10060633

**Published:** 2023-05-23

**Authors:** Khotibul Umam, Ching-Shan Feng, Greta Yang, Ping-Chen Tu, Chih-Yu Lin, Meng-Ting Yang, Tien-Fen Kuo, Wen-Chin Yang, Hieu Tran Nguyen Minh

**Affiliations:** 1Agricultural Biotechnology Research Center, Academia Sinica, Taipei 11529, Taiwan; khotibul.umam@uts.ac.id (K.U.); fchingshan@gmail.com (C.-S.F.); greyang900204@gmail.com (G.Y.); zelda@gate.sinica.edu.tw (C.-Y.L.); coral96reef@gmail.com (M.-T.Y.); tienfen@gate.sinica.edu.tw (T.-F.K.); 2Graduate Institute of Biotechnology, National Chung-Hsing University, Taichung 40227, Taiwan; 3Molecular and Biological Agricultural Sciences, Taiwan International Graduate Program, Academia Sinica, Taipei, Taiwan, and National Chung-Hsing University, Taichung 40227, Taiwan; 4Faculty of Life Science and Technology, Biotechnology Department, Sumbawa University of Technology, Sumbawa Besar 84371, NTB, Indonesia; 5Sun Ten Pharmaceutical Co., Ltd., New Taipei City 23143, Taiwan; pingchen.tu@gmail.com; 6Department of Life Sciences, National Chung-Hsing University, Taichung 40227, Taiwan; 7Graduate Institute of Integrated Medicine, China Medical University, Taichung 40402, Taiwan

**Keywords:** anti-bacterial activities, *Artemisia* plants, compound structure, mechanism, phytochemical

## Abstract

Over 70,000 people die of bacterial infections worldwide annually. Antibiotics have been liberally used to treat these diseases and, consequently, antibiotic resistance and drug ineffectiveness has been generated. In this environment, new anti-bacterial compounds are being urgently sought. Around 500 *Artemisia* species have been identified worldwide. Most species of this genus are aromatic and have multiple functions. Research into the *Artemisia* plants has expanded rapidly in recent years. Herein, we aim to update and summarize recent information about the phytochemistry, pharmacology and toxicology of the *Artemisia* plants. A literature search of articles published between 2003 to 2022 in PubMed, Google Scholar, Web of Science databases, and KNApSAcK metabolomics databases revealed that 20 *Artemisia* species and 75 compounds have been documented to possess anti-bacterial functions and multiple modes of action. We focus and discuss the progress in understanding the chemistry (structure and plant species source), anti-bacterial activities, and possible mechanisms of these phytochemicals. Mechanistic studies show that terpenoids, flavonoids, coumarins and others (miscellaneous group) were able to destroy cell walls and membranes in bacteria and interfere with DNA, proteins, enzymes and so on in bacteria. An overview of new anti-bacterial strategies using plant compounds and extracts is also provided.

## 1. Introduction

Since the 1940s, antibiotics have been widely used to treat bacterial infections in humans and animals. However, misuse and abuse of antibiotics have raised public health concerns about antimicrobial resistance (AMR), transmission of antibiotic resistance genes, ineffectiveness of current antibiotics, and the generation of superbugs. Over 700,000 people worldwide die of AMR each year, and by the year 2050, that number is projected to reach 10 million incurring medical expenses of more than 100 trillion USD [1]. There is thus an urgent need for measures to stop the spread of these disease-causing microbes. With COVID-19, combating AMR and preventing the emergence of new emerging drug-resistant organisms has become even more complex since 70% of COVID-infected patients rely on antibiotics for bacterial infections [2]. Therefore, searching for new anti-bacterial remedies is becoming an extremely important unmet need.

Eighty percent of the world population uses alternative medicine for their primary healthcare and most alternative medicine is derived from medicinal herbs [3]. Medicinal plants and compounds are an extraordinary source of drug leads and drugs [4]. Over 300,000 flowering plants have been identified. However, only 340,000 compounds of plant and other natural origins have been identified [5]. The WHO has listed 21,000 plants that are widely used for human medicinal purposes. These medicinal plants possess a variety of therapeutic activities that cover different categories of diseases. *Artemisia* species have been traditionally used as foods and medicines [6]. For example, *A. absinthium* has been found to be effective against bacterial infections, malaria, helminthes, leukemia, mental function, spasms, digestive diseases, diabetes, sclerosis, and cancers [7]. *A*. *argyi* is used as a traditional medicinal herb to treat amenorrhea, bruising, coagulation, dysmenorrhea, inflammation, microbial infections, jaundice, cancers, and metrorrhagia [8]. *A. afra* has been found effective against depression, cardiovascular disease, spasms, oxidation, and mycobacteria [9]. *A. annua* has been reported to be used to control fevers for over 2 millennia as well as malaria and bacterial infections [10]. *A. indica* had high anti-bacterial and antioxidant activities and, thus, its essential oil has been used in the pharmaceutical and food industries [11]. Although the *Artemisia* plants clearly possess a wide range of bioactivities, their antimicrobial activities and their anti-bacterial activity are of particular interest within the current context [4]. The anti-bacterial compounds in different *Artemisia* species have been explored, including the phytochemicals from *A. indica* [6], *A. argyi* [6], *A. annua* [12], *A. herba-alba* [12], and *A. feddei* [13]. In this review, we focus on the botany, medicinal uses, composition, function of anti-bacterial phytochemicals, and the modes of action of members of the *Artemisia* genus.

### 1.1. Botanical Properties and Traditional Use

*Artemisia* plants are annual and perennial herbs or subshrubs with a variety of leaf shapes, silvery or grayish foliage, and unshowy white or yellow flowers with tiny black seeds. In favorable conditions, they may reach a height of 80 to 150 cm. Over 500 species have been identified among this genus, which is the largest genus of the Asteraceae (daisy) family [14]. *Artemisia* plants belong to Plantae (Kingdom), Magnoliopsida (Class), Asterales (Order), Asteraceae (Family), and *Artemisia* (Genus) as described in Table 1. Due to their high diversity, authentication of the *Artemisia* species has been difficult and relies on molecular biology methods, chemotaxonomy as well as morphological characterization. The *Artemisia* plants are distributed in temperate, subtropical and tropical areas from low to high altitude worldwide [15].

*Artemisia* plants are aromatic plants due to their heavy scents that originate from essential oils, such as nootkatone in *A. annua* [16] and α-thujone (**29**) and β-thujone (**30**) in *A. absinthium* [17]. They are highly valued as food [18] and medicine [19]. The extracts and ingredients of the *Artemsia* plants are widely used in cosmetics, foods, drinks, and herbal medicines (moxibustion and decoction). This genus is also well-known for its antimicrobial uses. Artemisinin (**44**), a renowned anti-malarial prescription drug, is a terpenoid that was first identified from *A. annua* [20]. To date, around 1340 plants including the *Artemisia* plants have been claimed to possess anti-bacterial activities [21].

### 1.2. Chemical Compositions

The bioactivities of the *Artemisia* genus are attributable to their phytochemical composition [22]. As most *Artemisia* species are aromatic, they are rich in aromatic and volatile compounds though non-aromatic and non-volatile compounds exist. Basically, most of the identified compounds consist primarily of terpenoids, flavonoids, coumarins, and others (miscellaneous group). For instance, the major compounds in the *A. annua* essential oils were monoterpenoids including 30.7% artemisia ketone (**18**), 15.8% camphor (**33**), and 18.2% sesqueiterpenes [23]. Similarly, among 43 compounds identified from the aerial parts of *A. indica* [11], the essential oils included 42.1% artemisia ketone (**18**), 8.6% germacrene B (**16**), 6.1% borneol (**31**&**32**), and 4.8% (*Z*)-chrysanthenyl acetate (**27**). However, other *Artemisia* plants might have distinctive essential oils that make up their main components. For instance, *A. absinthium*, *A. herba-alba*, and *A. campestris*, have 32.07% β-pinene (**10**), 39.21% chamazulene, and 29.39% α-thujone (**29**), as major constituents of their essential oils, respectively [24]. The aerial parts of tarragon (*A. dracunculus*) oil was found to mainly be made up of 84% *p*-allylanisole (estragole) (**53**), 7.46% (*E*)-β ocimene (**3**), 6.24% (*Z*)-β ocimene (**4**), and 1.42% limonene (**6**&**7**). Similarly, the gas chromatography (GC) and mass spectroscopy (MS) analysis [25] indicated that the main essential oils of *A. argyi* included 16.2% 1,8-cineole (**28**), 14.3% β-pinene (**10**), 14% camphor (**33**), 13.9% artemisia ketone (**18**), and 11.1% α-pinene (**9**). Moreover, other laboratories reported that 40.33% of the *A. argyi* essential oils were caryophyllene oxide, neointermedeol, borneol (**31**&**32**), α-thujone (**29**) and β-caryophyllene (**12**) [26]. In the aerial parts of *A. vulgaris*, the principal essential oils included davanones (13.8 to 45.5%), germacrene D (**17**) (9.1 to 30.5%), 1,8-cineole (**28**) (16.4%), camphor (**33**) (18.9%), β-thujone (**30**) (8.9 to 10.9%), and (*Z*)-chrysanthenyl acetate (**27**) (10.4%) [27]. Therefore, *Artemisia* plants are rich in lipid-soluble components, especially in their essential oils. In addition, phytol (**46**), α-amyrin (**47**), betulinic acid (**48**), acacetin (**56**), 12α,4α-dihydroxybishopsolicepolide (**39**), and scopoletin (**61**) were isolated from *A. afra* [9].

Thanks to advances in GC and GC/MS methods, major and minor components in the essential oils of the different parts of *Artemisia* species can be identified. Pandey et al. (2017) found that the essential oils of the *Artemisia* plants were primarily from their aerial parts, followed by leaves, flowers, and buds. Furthermore, the main constituents of the *Artemisia* essential oils were artemisia ketone (**18**), camphene (**8**), β-pinene (**10**), caryophyllene (**12**), germacrene D (**17**), 1,8-cineole (**28**), thujones (**29**&**30**), and camphor (**33**) [28]. A total of 75 compounds, including 49 terpenoids, 11 flavonoids, 2 coumarins, and 13 other compounds are delineated in Table 2, Table 3 and Table 4.

### 1.3. Classification of Anti-Bacterial Phytochemicals

To seek phytochemicals present in the *Artemisia* plants, we used key words to seek the compounds in KNApSAcK metabolomics databases [29] in cross-reference with literature in Pubmed, Google scholar, and Web of Science, as shown in Figure 1. We discovered 946 compounds, including 912 compounds from the KNApSAcK metabolomics databases and 32 compounds from the literature search. These compounds were from 122 known *Artemisia* species and 24 unidentified *Artemisia* spp. To narrow down 946 compounds to those with anti-bacterial functions, we cross-referenced the compounds with text search in Pubmed, Google scholar, and Web of Science database using the key words, “anti-bacterial”, and “antimicrobial”. Consequently, 75 compounds with anti-bacterial activities were selected, curated, and finalized as described in Figure 1. These compounds from the *Artemisia* species were classified into three groups based on their chemical structures (Table 2, Table 3 and Table 4), terpenes and terpenoids (Table 2), polyphenol (Table 3), and a miscellaneous (other) group (Table 4). The information about the structure, molecular weight, bacteria, anti-bacterial activity, and plant species of the 75 compounds is appended in each Table.

#### 1.3.1. Terpenes and Terpenoids

Forty-nine terpenes and terpenoids with anti-bacterial properties found in *Artemisia* plants are listed in Table 2. They constitute the majority of compounds in *Artemisia* as described in Section 1.2 (Chemical composition). Terpenes, a simple hydrocarbons structures, while terpenoids (oxygen-containing hydrocarbons) are defined as modified class of terpenes with various functional groups and oxidized methyl groups moved or deleted at various places which classified into alcohols, ethers, aldehydes, phenols ketones, esters, and epoxides that are volatile [30,31]. Terpenes contain ten monoterpenes and seven sesquiterpenes. There are also compounds identified as terpenoids, consisting of sixteen monoterpenoids, twelve sesquiterpenoids, one diterpenoids, and three triterpenoids as listed in Table 2.

**Table 2 bioengineering-10-00633-t002:** Classification, structure, and anti-bacterial properties of terpenes and terpenoids from the *Artemisia* species.

SN ^a^	Name	Structure	MW	Pathogen	MIC (µg/mL)	Plant
Terpenes (monoterpenes)
**1**	Sabinene	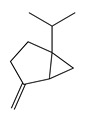	154.3	*B. subtilis*, *S. epidermidis*	>64 [11]	*A. indica* [11]
*P. aeruginosa*	>32 [11]
*S. aureus*	32 [11]
*S. typhi*	128 [11]
*S. dyssenteriae*	>128 [11]
*K. pneumonia*	64 [11]
**2**	Myrcene	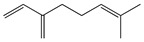	136.2	*S. epidermidis*	121 [32]	*A. absinthium* [7]
*B. subtillis*	322.11 [32]
*S. dyssenteriae*	325 [32]
*K. pneumonia*	400 [32]
**3**	(*E*)-β-Ocimene	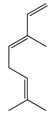	136.2	*B. subtillis*, *S. epidermidis*,	130 [32]	*A. dracunculus* [33]
*P. vulgaris*	220 [32]
*S. dyssenteriae*	650 [32]
*K. pneumonia*	600 [32]
**4**	(*Z*)-β-Ocimene	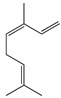	136.2	*B. subtillis*	130 [32]	*A.dracunculus* [33]
*S. dyssenteriae*	220 [32]
*E. coli*, *K. pneumonia*	600 [32]
**5**	*p*-Cymene	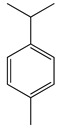	134.2	*B. subtilis, S. typhi*	>64 [11]	*A. indica* [11]
*S. epidermidis*	128 [11]
*P. aeruginosa*	64 [11]
*S. aureus*	32 [11]
*S. dyssenteriae*	>128 [11]
*K. pneumonia*	>64 [11]
**6**	(+)-Limonene	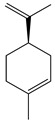	136.2	*L. monocytogenes*	20 [34]	*A. capillaris* [35]
*E. coli* ^cs^	112 [35]
*H. influenzae*	128 [35]
Methicillin-resistant *S. aureus^cs^* (MRSA^cs^)	150 [35]
**7**	(−)-Limonene	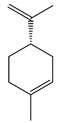	136.2	*S. pyogenes*, *K. pneumoniae*	156 [35]
*S. pneumoniae*	198 [35]
MRSA	332 [35]
MRSA	330 [35]
**8**	Camphene	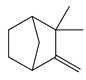	136.2	*V. vulnificus*	400 [36]	*A. iwayomogi* [36]
*S. aureus*, *S. mutans*, *E. coli* ATCC25922, *C. freundii*	1600 [36]
*S. epidermidis*	3200 [36]
*S. pyogenes*, *E. faecalis*, *E. gallinarum*, *S. typhimurium*, *E. coli* O157:H7, *E. cloacae*, *P. aeruginosa*	>12,800 [36]
*K. pneumonia*	64 [11]	*A. indica* [11]
*P. aeruginosa*	128 [11]
*B. subtilis*, *S. epidermidis, S. typhi*	>128 [11]
*S. dyssenteriae*	256 [11]
**9**	α-Pinene	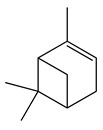	136.2	*E. coli* ^cs^	98 [37]	*A. vestita* [37]
*H. influenzae*	126 [37]
*S. pyogenes*	132 [37]
MRSA^cs^, *S. pneumoniae*	172 [37]
*K. pneumoniae*	178 [37]
MRSA	210 [37]
Methicillin and gentamycin-resistant *S. aureus* (MGRSA), *S. aureus*	256 [37]
*S. aureus*, *K. pneumonia*	32 [11]	*A. indica* [11]
*P. aeruginosa*	64 [11]
*B. subtilis*, *S. typhi*, *S. dyssenteriae*	128 [11]
*S. epidermidis*	>128 [11]
**10**	β-Pinene	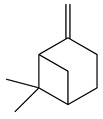	136.2	*E. coli* ^cs^	102 [37]	*A. vestita* [37]
*H. influenzae*	132 [37]
*S. pyogenes*	144 [37]
MRSA^cs^, *S. pneumoniae*, *K. pneumoniae*	170 [37]
MRSA	210 [37]
MGRSA	256 [37]
*S. aureus*	32 [11]	*A. indica* [11]
*K. pneumonia*	>32 [11]
*P. aeruginosa*	64 [11]
*B. subtilis*, *S. typhi*	>64 [11]
*S. epidermidis*, *S. dyssenteriae*	128 [11]
Terpenes (sesquiterpenes)
**11**	α-Elemene	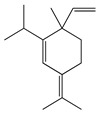	204.3	*S. enterica*	0.1 [6]	*A. indica* [6], *A. dracunculus* [38]
*E. coli*	25 [6]
S. *typhimurium*	60 [38]
*B. cereus*, *S. aureus*	250 [38]
*L. monocytogenes*	262.5 [6]
**12**	β-Caryophyllene	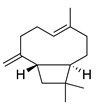	204.3	*S. pyogenes*	25 [36]	*A. capillaris* [35], *A. iwayomogi* [36], *A. feddei* [13], *A. argyi* [26]
*S. aureus*, *E. gallinarum*	50 [36]
*H. influenzae*, *K. pneumoniae*	64 [35]
*E. coli* ^cs^	92 [35]
*S. epidermidis*	100 [36]
*S. pneumoniae*	122 [35]
*S. pyogenes*	126 [35]
MRSA^cs^	144 [35]
*E. faecalis*	200 [36]
MRSA	330 [35]
MGRSA	332 [35]
*P. gingivalis*	400 [13]
*S. mutans* [36], *P. intermedia* [13]	800
*S. mutans*, *S. sanguinis*, *S. gordonii*, *A. actinomycetemcomitans*	1600 [13]
*V. vulnificus*	6400 [36]
*E. coli*, *S. aureus*, *S. epidermidis*, *S. pyogenes*, *S. sobrinus*, *S. ratti*, *S. criceti*, *S. anginosus*, *F. nucleatum*	12,800 [13]
*S. typhimurium*, *E. coli* ATCC25922, *E. coli* O157:H7, *E. cloacae*, *P. aeruginosa*, *C. freundii*	>12,800 [36]
**13**	α-Farnesene	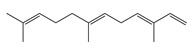	204.3	*S. enterica*	3.12 [6]	*A. indica* [6]
*E. coli*	200 [6]
*L. monocytogenes*	4000 [6]
**14**	α-Curcumene	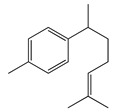	202.3	*E. coli*	100 [39]	*A. integrifolia* L. [39]
*S. aureus*	130 [39]
*B. cereus*	140 [39]
*Y. enterocolitica*	260 [39]
**15**	Dihydro-*ar*-curcumene	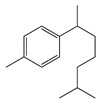	204.4	*E. coli*	120 [39]	*A. integrifolia* L. [39]
*E. coli* ATCC25922	200 [6]
*Y. enterocolitica*	280 [39]
*L. monocytogenes*	4000 [6]
**16**	Germacrene B	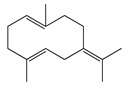	204.4	*S. aureus*	32 [11]	*A. indica* [11]
*P. aeruginosa*, *S. typhi*	>32 [11]
*K. pneumonia*	64 [11]
*B. subtilis,* *S. epidermidis*	>64 [11]
*S. dyssenteriae*	>128 [11]
**17**	Germacrene D	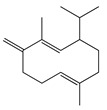	204.4	*B. subtillis*	30.3 [32]	*A. vulgaris, A. annua, A. herba*-*alba* [28]
*S. aureus*	30.3 [32]
*P. vulgaris, S. dyssenteriae*	65.1 [32]
*K. pneumonia*	90.1 [32]
*S. typhi*	90.2 [32]
Terpenoids (monoterpenoids)
**18**	Artemisia ketone	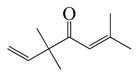	152.2	*S. aureus*	>16 [11]	*A. indica* [11]
*P. aeruginosa*, *S. typhi*	32 [11]
*K. pneumonia*	>32 [11]
*B. subtilis*	64 [11]
*S. epidermidis*	>64 [11]
*S. dyssenteriae*	>128 [11]
**19**	Linalool	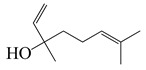	154.3	*B. cereus*, *E. coli*, *S. aureus*, *S. typhimurium*	250 [38]	*A. annua* [10], *A. dracunculus* [38]
**20**	Nerol	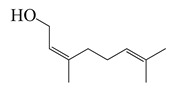	154.3	*S. epidermidis*	>16 [11]	*A. indica* [11]
*S. aureus*	32 [11]
*P. aeruginosa*	64 [11]
*B. subtilis*, *K. pneumonia*, *S. typhi*	>64 [11]
*S. dyssenteriae*	128 [11]
**21**	Grandisol	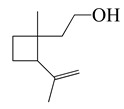	154.2	*S. pyogenes*, *H. influenzae*	130 [37]	*A. vestita* [37]
*S. pneumoniae*	132 [37]
*K. pneumoniae*	144 [37]
MRSA	178 [37]
**22**	Piperitone	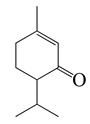	152.2	*H. influenzae, E. coli* ^cs^	72 [37]	*A. vestita* [37]
*K. pneumoniae*	86 [37]
*S. pyogenes*	102 [37]
MRSA, *S. pneumoniae*	112 [37]
MRSA^cs^	122 [37]
MGRSA	156 [37]
**23**	Terpinen-4-ol	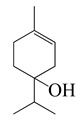	154.2	*S. gordonii*	50 [13]	*A. feddei* [13]
*F. nucleatum*, *P. intermedia*	200 [13]
*P. gingivalis*	400 [13]
*E. coli*, *S. pyogenes*, *S. aureus*, *S. ratti*, *S. anginosus*, *S. epidermidis*, *S. mutans*, *S. sanguinis*, *S. sobrinus*, *A. actinomycetemcomitans*	1600 [13]
*S. criceti*	3200 [13]
**24**	α-Terpineol	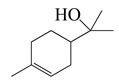	154.2	*S. aureus*	30 [40]	*A. feddei* [13], *A. princeps* Pamp. [41]
*S. gordonii*	50 [13]
*E. coli*	60 [40]
*B. cereus*, *S. Typhimurium*	120 [40]
*P. intermedia*, *P. gingivalis*	200 [13]
*F. nucleatum*	400 [13]
*G. vaginalis*	560 [41]
*S. aureus*, *S. epidermidis*, *S. pyogenes*, *E. coli*, *S. mutans*, *S. sobrinus*, *S. ratti*, *S. anginosus, S. sanguinis*, *A. actinomycetemcomitans*	1600 [13]
*S. criceti*	3200 [13]
**25**	Thymol	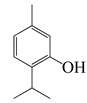	150.2	*S. aureus*, *K. pneumoniae* K38^cs^	60 [42]	*A. haussknechtii* [42]
*E. coli, P. aeruginosa*, *A. baumannii* A52^cs^	80 [42]
**26**	Carveol	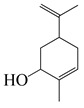	152.2	*S. enterica*	0.1 [6]	*A. indica* [6], *A. dracunculus* [38]
*S. aureus*	15 [38]
*E. coli*	25 [6]
*S. typhimurium*	30 [38]
*E. coli*	60 [38]
*L. monocytogenes*	100 [6]
*B. cereus*	120 [38]
**27**	(*Z*)-Chrysanthenyl acetate	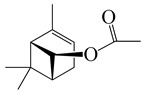	194.3	*B. subtilis*, *S. typhi*	256 [11]	*A. indica* [11]
*S. aureus*	512 [11]
*K. pneumonia*	>256 [11]
*S. dyssenteriae*	>128 [11]
**28**	1,8-Cineole	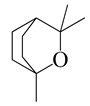	154.3	*H. influenzae*	98 [37]	*A.feddei* [13], *A. vestita* [37], *A. iwayomogi* [36]
*K. pneumoniae*, *E. coli*^cs^	102 [37]
*S. pyogenes*	112 [37]
*S. pneumoniae*	132 [37]
MRSA^cs^	152 [37]
MRSA	244 [37]
MGRSA	256 [37]
*S. epidermidis*	800 [13]
*P. intermedia*	1600 [13]
*S. pyogenes*, *E. coli*, *V. vulnificus* [36], *S. anginosus*, *F. nucleatum* [13]	3200 [13]
*S. aureus*, *S. mutans*, *E. faecalis*, *E. gallinarum*, *S. typhimurium*, *S. epidermidis*, *E. coli* O157:H7, *E. cloacae*, *C. freundii* [36], *S. gordonii*, *A. actinomycetemcomitans*, *P. gingivalis* [13]	6400 [13]
*S. aureus*, *S. pyogenes*, *S. sanguinis*, *S. mutans*, *S. ratti*, *S. sobrinus*, *S. criceti*	12,800 [13]
*P. aeruginosa*	>12,800 [36]
*S. enterica*	0.1 [6]	*A. indica* [6]
*E. coli*	25 [6]
*L. monocytogenes*	100 [6]
**29**	α-Thujone	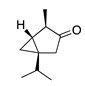	152.2	*S. aureus*	32 [11]	*A. indica* [11], *A. absinthium* [17]
*K. pneumonia*	>32 [11]
*P. aeruginosa*	64 [11]
*B. subtilis*, *S. epidermidis*	>64 [11]
*S. typhi*, *S. dyssenteriae*	128 [11]
**30**	β-Thujone	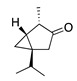	152.2	*S. aureus*	90 [43]	*A. indica* [11], *A. absinthium* [17]
*S. epidermidis*	100 [43]
*E. coli*	350 [43]
*K. pneumoniae*	650 [43]
*P. aeruginosa*	750 [43]
*E. cloacae*	830 [43]
**31**	(+)-Borneol	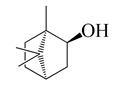	154.3	*S. aureus* CCARM3523, *S. typhimurium* KCCM11862	2 [44]	*A. iwayomogi* [44], *A. feddei* [13]
*S. aureus* strains CCARM0027 & CCARM3511, *S. typhimurium* CCARM 8007	4 [44]
*S. typhimurium* CCARM8009	8 [44]
*S. enteritidis* strains KCCM12201, CCARM8010 & CCARM8011	>16 [44]
*S. aureus*	30 [40]
*B. cereus*, *S. typhimurium*	120 [40]
*V. vulnificus*	100 [14]
*F. nucleatum*, *P. intermedia*	200 [13]
*E. coli*	250 [40]
*S. pyogenes*, *E. coli* O157:H7 [14], *E. coli*, *S. sobrinus* [13]	400
*S. epidermidis*, *S. pyogenes*, *S. mutans*, *S. anginosus*, *S. gordonii*, *A. actinomycetemcomitans*, *P. gingivalis*	800 [13]
*E. faecalis*, *E. gallinarum* [14], *S. aureus*, *S. sanguinis* [13]	1600
*S. ratti*, *S. criceti*	3200 [13]
*P. aeruginosa*	>12,800 [14]
**32**	(−)-Borneol	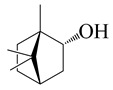	154.3	*B. cereus*, *E. coli*, *S. aureus*	250 [40]	*A. argyi*[26], *A. indica* [11]
*S. typhimurium*	800,000 [40]
*B. subtilis,*	128 [11]
*S. epidermidis*	64 [11]
*P. aeruginosa*, *S. typhi*	>64 [11]
*S. dyssenteriae*	>128 [11]
*K. pneumonia*	64 [11]
**33**	Camphor	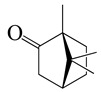	152.2	*S. aureus* CCARM0027 & CCARM3523, *S. typhimurium* KCCM11862	2 [44]	*A. annua* [10,14]
*S. aureus* CCARM3511, *S. typhimurium* CCARM 8007 & CCARM8009	4 [44]
*S. aureus*	15 [40]
*S. enteritidis* KCCM12201, CCARM8010 & CCARM8011	>16 [44]
*B. cereus*, *S. typhimurium*, *E. coli*	250 [40]
*V. vulnificus*	400 [14]
*C. perfringens*	500 [10]
*S. epidermidis*, *S. pyogenes*, *E. coli* O157:H7	800 [14]
*S. aureus*, *S. mutans*, *E. coli*, *E. cloacae*, *C. freundii*	1600 [14]
*E. faecalis*, *E. gallinarum*	3200 [14]
*S. typhimurium*	6400 [14]
*P. aeruginosa*	>12,800 [14]
Terpenoids (sesquiterpenoids)
**34**	Vulgarone B	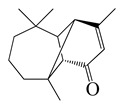	218.3	*S. aureus* CCARM0027	0.5 [44]	*A. iwayomogi* [44]
*S. aureus* CCARM3511 & CCARM3523, *S. typhimurium* KCCM11862	1 [44]
*S. typhimurium*	2 [44]
*S. enteritidis* KCCM12201, CCARM8010 & CCARM8011, *S. typhimurium* CCARM 8007	>2 [44]
**35**	(+)-(*S*)-*ar*-Turmerone	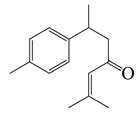	217.3	*B. cereus*	100 [39]	*A. integrifolia* [39]
*E. coli*	120 [39]
*S. aureus*	150 [39]
*Y. enterocolitica*	180 [39]
**36**	(+)-(*S*)-Dihydro-*ar*-turmerone	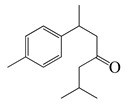	219.4	*B. cereus*, *Y. enterocolitica*	120 [39]	*A. integrifolia* [39]
*E. coli*	140 [39]
*S. aureus*	160 [39]
**37**	Zerumbone	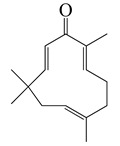	218.3	*E. coli*	70 [39]	*A. integrifolia* [39]
*B. cereus*	90 [39]
*S. aureus*	110 [39]
*Y. enterocolitica*	230 [39]
**38**	Dehydroleucodine	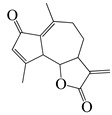	244	*H. pylori* HP786^cs^	1 [45]	*A. douglasiana* [45]
*H. pylori*^cs^, NCTC 11,638 & H796	2 [45]
*H. pylori*^cs^ HP781 & HP795	4 [45]
*H. pylori*^cs^ HP788 & HP789	8 [45]
**39**	12α,4α-Dihydroxybishopsolicepolide	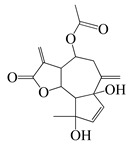	320.3	*A. naeslundii*, *A. israelii*	0.5 [9]	*A. afra* [9]
*P. intermedia*	1 [9]
*A. actinomycetemcomitans*, *P. gingivalis*	>1 [9]
**40**	1,3,8-Trihydroxyeudesm-4-en-7α,11β*H*-12,6α-olide	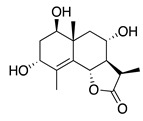	282	*B. subtilis*	25 µg/5 µL DMSO/disc [46]	*A. herba-alba* [46]
*S. aureus*	50 µg/5 µL DMSO/disc [46]
**41**	3α,8β-Dihydroxygermacr-4(15),9(10)-dien-7β,11α*H*,12,6α-olide	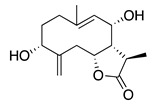	266.3	*B. subtilis*, *S. aureus*	100 µg/5 µL DMSO/disc [46]	*A. herba-alba* [46]
*E. coli*	50 µg/5 µL DMSO/disc [46]
**42**	1β,8α-*D*ihydroxy-11α,13-dihydrobalchanin	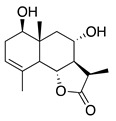	264.3	*S. aureus, B. subtilis*	100 µg/5 µL DMSO/disc [46]	*A. herba-alba* [46]
*E. coli*	50 µg/5 µL DMSO/disc [46]
**43**	11-Epiartapshin	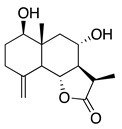	264.3	*S. aureus*	25 µg/5 µL DMSO/disc [46]	*A. herba-alba* [46]
*E. coli*	50 µg/5 µL DMSO/disc [46]
*B. subtilis*	100 µg/5 µL DMSO/disc [46]
**44**	Artemisinin	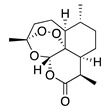	282	*S. aureus*, *B. subtilis*, *Salmonella* sp.	90 [47]	*A. annua* [47]
**45**	Artesunate	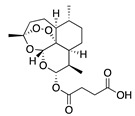	384.4	MRSA	>4096 [48]	*A. annua* [49]
Terpenoids (diterpenoids)
**46**	Phytol	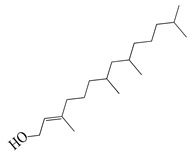	296.5	*A. israelii*	0.3 [9]	*A. afra* [9]
*A. naeslundii*, *P. intermedia*	1 [9]
*A. actinomycetemcomitans*	>1 [9]
Terpenoids (triterpenoids)
**47**	α-Amyrin	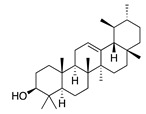	426.7	*A. naeslundii*	1 [9]	*A. afra* [9]
*A. israelii*, *A. actinomycetemcomitans*, *P. intermedia*, *P. gingivalis*	>1 [9]
**48**	Betulinic acid	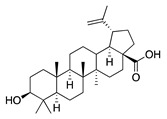	456.7	*A. naeslundii*	0.3 [9]	*A. afra* [9]
*A. israelii*, *P. intermedia*, *P. gingivalis*	1 [9]
*A. actinomycetemcomitans*	>1 [9]
**49**	Ursolic acid	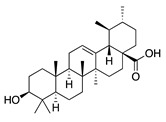	456.7	*S. aureus* ATCC6538	32 [50]	*A. annua* [49]
*E. coli* ATCC25922, *K. pneumoniae*, *S. flexneri*	64 [50]
*E. coli* ATCC27, *P. aeruginosa*	512 [50]
*S. aureus* ATCC12692 & ATCC12624, *V. colareae*, *L. monocytogenes*, *B. cereus*, *A. caveae*	≥1024 [50]

^a^ SN: serial number, ^CS^ clinical strain.

#### 1.3.2. Polyphenols

In *Artemisia*, there are 13 anti-bacterial compounds, including 11 flavonoids and 2 coumarins, as listed in Table 3.

**Table 3 bioengineering-10-00633-t003:** Classification, structure, and anti-bacterial properties of polyphenols from the *Artemisia* species.

SN ^a^	Name		Structure	MW	Pathogen	MIC (µg/mL)		Plant
Flavonoids								
**50**		Acacetin	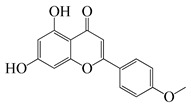	284.3	*A. israelii*	0.3 [9]	*A. afra* [9]	
					*A. naeslundii*, *P. intermedia*, *P. gingivalis*	1 [9]		
					*A. actinomycetemcomitans*	>1 [9]		
**51**		Casticin	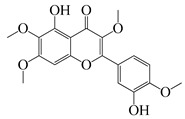	374.3	*C. perfringens* 200302-1-1-Ba	800 [10]	*A. annua* [10]	
**52**		Chrysosplenetin	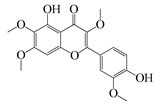	374.3	*S. aureus*	-	*A. rupestris* [51]	
**53**		Chrysoeriol	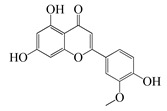	300.2	*S. aureus*	-	*A. rupestris* [51]	
**54**		Chrysosplenol B	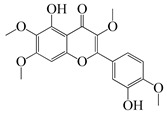	374.3	*E. coli* WT, *E. coli* Δ*tolC*, *E. coli* Δ*tolC* (*fabl*)	>37.4 [52]	*A. californica* [52]	
**55**		Chrysosplenol D	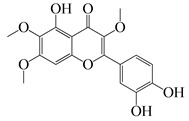	360.3	*C. perfringens* 200302-1-1-Ba	200–400 [10]	*A. annua* [10]	
					*B. subtilis*, *E. coli*, *P. fluorescens,* and *M. tetragenus*	250–500 [10]		
**56**		Penduletin	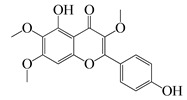	344.3	*S. aureus*	-	*A. rupestris* [51]	
**57**		Artemetin	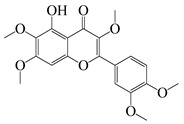	388.4	MRSA	-	*A. rupestris* [51]	
**58**		Pachypodol	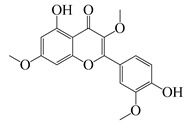	344.3	MRSA	-	*A. rupestris* [51]	
**59**		Jaceosidin	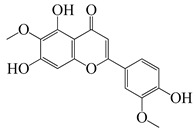	330	*E. coli* Δ*tolC*, *E. coli* Δ*tolC* (*fabl*)	3.3 [52]	*A. californica* [52], *A. argyi* [53]	
					*E. coli* WT	>33 [52]		
**60**		Jaceidin	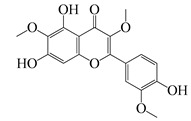	360	*E. coli* Δ*tolC*, *E. coli* Δ*tolC* (*fabl*)	18 [52]	*A. californica* [52]	
					*E. coli* WT	>36 [52]		
Coumarins								
**61**		Scopoletin	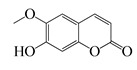	192.2	*A. israelii*	0.3 [9]	*A. afra* [9]	
					*P. intermedia*	0.5 [9]		
					*A. naeslundii*	1 [9]		
					*A. actinomycetemcomitans*, *P. gingivalis*	>1 [9]		
**62**		5-β-D-Glucopyranosyloxy-7-methoxy-6*H*-benzopyran-2-one	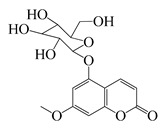	354	*B. subtilis*	25 µg/5 µL DMSO/disc [46]	*A. herba-alba* [46]	
					*S. aureus*, *E. coli*	100 µg/5 µL DMSO/disc [46]		

^a^ SN: serial number, ^CS^ clinical strain.

#### 1.3.3. Miscellaneous Group

This group contains the thirteen compounds that are absent from the first three groups. Their chemical structure, molecular weight, anti-bacterial properties, and plant species are tabulated below.

**Table 4 bioengineering-10-00633-t004:** Classification, structure, and anti-bacterial properties of miscellaneous compounds from the *Artemisia* species.

SN ^a^	Name	Structure	MW	Pathogen	MIC (µg/mL)	Plant
**63**	2,4-Dihydroxy-6-methoxyacetophenone	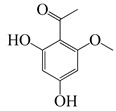	182.2	*C. perfringens* 200302-1-1-Ba	800 [10]	*A. annua* [10]
**64**	Capillin	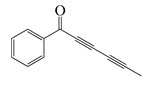	168.2	*S. pneumoniae*, *H. influenzae*, *E. coli*^cs^	64 [35]	*A. capillaris* [35]
				*K. pneumoniae*	72 [35]	
				*S. pyogenes*	98 [35]	
				MRSA^cs^	112 [35]	
				MRSA, MGRSA	156 [35]	
**65**	Benzoic acid *p*-(β-D-glucopyranosyloxy)-methyl ester	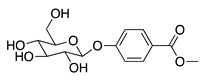	312.3	*B. subtilis*	50 µg/5 µL DMSO/disc [46]	*A. herba-alba* [46]
				*S. aureus*	25 µg/5 µL DMSO/disc [46]	
**66**	Diisooctyl phthalate	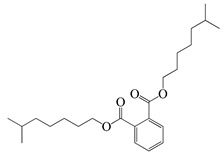	390.6	*S. enterica*	1.56 [6]	*A. indica* [6]
				*E. coli*	200 [6]	
				*L. monocytogenes*	30,000 [6]	
**67**	Integracid	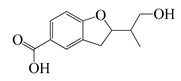	221	*S. aureus*	60 [54]	*A. integrifolia* [39]
				*B. cereus*	80 [39]	
				*E. coli*	100 [39]	
				*Y. enterocolitica*	120 [39]	
**68**	(*Z*)-2-(Hexa-2,4-diyn-1-ylidene)-1,6-dioxaspiro [4.5]dec-3-ene	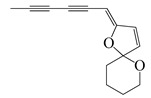	124.3	*S. aureus*	4.7 [55]	*A. pallens* [55]
				*B. subtilis*	7.8 [55]	
				*E. coli*	8.4 [55]	
				*P. aeruginosa*	10 [55]	
**69**	(*E*)-2-(Hexa-2,4-diyn-1-ylidene)-1,6-dioxaspiro [4.5]dec-3-ene	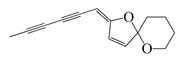	124.3	*S. aureus*	1.6 [55]	*A. pallens* [55]
				*E. coli*	2.1 [55]	
				*B. subtilis*	2.4 [55]	
				*P. aeruginosa*	2.7 [55]	
**70**	Ponticaepoxide	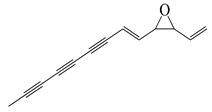	182.2	*C. perfringens* 200302-1-1-Ba	100–200 [10]	*A. annua* [10]
**71**	(+)-*threo*-(5*E*)-Trideca-1,5-dien-7,9,11-triyne-3,4-diol	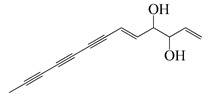	200.3	*C. perfringens* 200302-1-1-Ba	400–800 [10]	*A. annua* [10]
**72**	Methyl linolenate	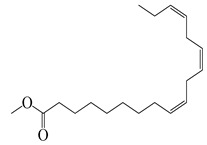	292.5	*S. enterica*	0.19 [6]	*A. indica* [6]
				*E. coli*	25 [6]	
				*L. monocytogenes*	6400 [6]	
**73**	Estragole	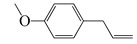	148.2	*S. aureus* 1199	101.6 [56]	*A. annua* [10]
				*S. aureus* 1199B	128 [56]	
**74**	Rosmarinic acid (RA)	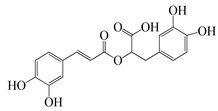	360	*S. aureus*, *E. coli*	500 [57]	*A. absinthium*, *A. annua*, *A. alba*, etc. [58]
				*S. aureus*	800 [59]	
				MRSA^cs^	10,000 [59]	
**75**	Eugenol	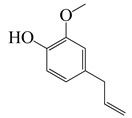	164.2	carbapenem-resistant *Klebsiella pneumoniae* (CRKP)	200 [60]	*A. annua* [10]

^a^ SN: serial number, ^CS^ clinical strain.

## 2. Anti-Bacterial Properties of the *Artemisia* Plants

Since the *Artemisia* plants have been used medicinally for bacterial and other infections, their extracts and compounds are often assessed in vitro and in animal models against typical harmful zoonotic bacteria such as methicillin-resistant *S. aureus* (MRSA), *E. coli*, *S. typhimurium*, *S. aureus*, etc. The inhibitory effects of the extracts and compounds of the *Artemisia* plants on bacteria are generally assessed based on serial dilution, agar plate assay, and/or disc diffusion methods as used in Table 2, Table 3, Table 4. Their minimum inhibitory concentration (MIC) is used if applicable. Table 2, Table 3, Table 4 list the compounds with MIC values against specific bacteria. The anti-bacterial activity of the *Artemisia* species and compounds can be ranked based on the MIC value [61] as described in Appendix A.

Using 0.2 mL of each extract of three different *Artemisia* species in the agar diffusion method, Poiată et al. discovered that *A. annua* methanolic and ethanolic extracts are most active in suppressing five Gram-positive bacteria (MRSA, *S. aureus* ATCC 25923, *S. lutea* ATCC 14579, and *B. subtilis* ATCC 6633) and a Gram-negative *P. aurugi* [62]. Furthermore, the next active extract was a methanolic extract of *A. absinthium*, which inhibited MRSA, *S. lutea, B. subtilis*, and *P. auruginosa*. The ethanolic extracts of *A. vulgaris*, and *A. absinthium,* later *A. annua* hexane extract and *A. vulgaris* ethanolic extract, inhibited only one to three pathogens, while no anti-bacterial activity was observed in *A. absinthium* and *A. vulgaris* hexane extracts [62]. The results of these investigations conclude that anti-bacterial activity can differ among various *Artemisia* species. This difference could be ascribed to the composition of their phytochemicals.

We were able to categorize our listed compounds into different levels of anti-bacterial activity using Appendix A. *A. annua* is the most studied plant in the *Artemisia* genus. Ivarsen et al. reported that dichloromethane and n-hexane extracts of the aerial parts of *A. annua* were active against *C. perfringens* with a MIC value of 185 and 270 µg/mL, respectively [10]. In addition, *A. annua* extracts have also been reported to inhibit a significant number of pathogens such as *E. coli*, *S. typhi*, *B. subtilis*, *S. aureus*, and *C. perfringens* [63,64]. The high anti-bacterial properties of *A. annua* are due to its primary compounds from essential oil including monoterpenoids group were 30.7% artemisia ketone **(18)**, 15.8% camphor **(33)**, and 18.2% from sesqueiterpenes [23] and also from sabinene (**1**), linalool (**19**), camphene (**8**), α-pinene (**9**), α-terpineol (**24**), borneol (**31**&**32**), camphor (**33**), eugenol (**75**), and coumarins [10]. The terpenoids (monoterpenoids) α-terpineol (**24**) is also found in *A. feddei* [13] and *A. princeps* Pamp [41] with a very low to very high activity against 20 pathogenic bacteria with a MIC value of 30 to 3200 µg/mL [13,41]. Likewise, borneol (**31**&**32**), commonly present in *A. feddei*, *A. indica*, *A. argyi*, and *A. iwayomogi*, showed a MIC value of 2 to 12,800 µg/mL against 44 zoonotic bacteria such as *S. aureus*, *S. typhimurium*, *S. enteritidis*, *B. cereus*, *V. vulnificus*, *F. nucleatum*, *P. intermedia*, *S. pyogenes*, *E. coli*, *S. sobrinus*, *S. epidermidis*, *S. pyogenes*, *S. mutans*, *S. anginosus*, *S. gordonii*, *A. actinomycetemcomitans*, *P. gingivalis*, *E. faecalis*, *E. gallinarum*, *S. sanguinis*, *S. ratti*, *S. criceti*, *P. aeruginosa*, *K. pneumonia*, and *S. dyssenteriae* [11,13,14,44]. Using a bioactivity-directed fractionation and isolation strategy, β-caryophyllene (**12**) was identified and this compound inhibited 36 pathogenic and clinical bacteria, particularly *S. pyogenes*, *S. aureus*, *E. gallinarum*, *H. influenzae*, *K. pneumonia*, MRSA, methicillin and gentamycin-resistant *S. aureus* (MGRSA), etc. with a MIC value ranging from 25 to over 12,800 µg/mL [13,35,36]. Furthermore, it is possible that most terpenoids have greater anti-bacterial activity than terpenes. The MIC value of one terpenoids (monoterpenoids), camphor (**33**)**,** and one triterpenoid, ursolic acid (**49**), towards different bacteria was 2 to 1280 µg/mL [14] and 32 to 1280 µg/mL [50], respectively. Accordingly, a terpenoids (sesquiterpenoids), artemisinin (**44**) (MIC = 90 µg/mL), had a higher activity than a monoterpenoid, linalool (**19**) (MIC 250 µg/mL) against *B. cereus*, *E. coli*, *S. aureus*, and *S. typhimurium* in *A. annua* [38]. One monoterpene, myrcene (**2**), presents in the *A. absinthium* extract, indicated very low-moderate activities against *S. epidermidis*, *B. subtillis*, *S. dyssenteriae*, and *K. pneumonia* with a MIC value of 121, 322.11, 325, and 400 µg/mL, respectively [32]. Similarly, α-thujone (**29**) and β-thujone (**30**) suppressed *S. aureus*, *S. epidermidis*, *E. coli*, *K. pneumoniae*, *P. aeruginosa*, and *E. cloacae* with MIC values of 90, 100, 350, 650, 750, and 830 µg/mL, respectively [11,43].

As shown in Table 4, ponticaepoxide (**70**), (MIC = 100–200 µg/mL), identified from *A. annua*, had a higher anti-bacterial activity than (+)-*threo*-(5*E*)-trideca-1,5-dien-7,9,11-triyne-3,4-diol (**71**) (MIC = 400–800 µg/mL) [10]. Terpinen-4-ol (**23**) of *A. feddei* had a MIC of 50–3200 µg/mL against 15 pathogens [13]. A phenylpropanoid, estragole (**73**) showed a higher activity towards *S. aureus* species with a MIC value of 101.6 to128 µg/mL [56] than that (MIC = 200 µg/mL) of eugenol (**75**), [60]. As listed in Table 3, flavonoids possess low anti-bacterial activities. For instance, chrysosplenol D (**55**), a major flavonoid found in *A. annua* extracts, displayed a very low to low anti-bacterial activity against *C. perfringens*, *E. coli*, *B. subtilis*, *M. tetragenus*, and *P. fluorescens* with a MIC value of 200 to 500 µg/mL [10]. Moreover, the other flavonoids, casticin (**51**) showed lower anti-bacterial activity against *C. perfringens* 200302-1-1-Ba with a MIC value of 800 µg/mL [10]. Moreover, a popular phenolic, rosmarinic acid (RA) (**74**), showed very low anti-bacterial activity against *S. aureus*, *E. coli*, and MRSA^cs^ with a MIC of 500 to 10,000 µg/mL [57,59]. This compound was found in *A. absinthium*, *A. annua*, *A. alba*, and so on [58].

As mentioned in Section 1.2, artemisia ketone (**18**), germacrene B (**16**) (MIC = 32–128 µg/mL), and borneol (**33**&**34**) (MIC = 32–128 µg/mL) were discovered to exhibit moderate to very high anti-bacterial activities. In *A. indica* found that the essential oil of artemisia ketone (**18**), germacrene B (**16**), and borneol (**31**&**32**) were 42.1%, 8.6%, and 6.1% respectively. In contrast, (*Z*)-chrysanthenyl acetate (**27**) (MIC value of 128–512 µg/mL) with 4.8% essential oil had very low to moderate anti-bacterial activities against 7 clinical bacteria [11]. Similar to the bacterial efficacy of artemisia ketone (**18**), sabinene (**1**), and *p*-cymene (**5**) at MICs ranging 32 to >128 µg/mL in the same test [11]. Interestingly, among the terpenoids in *A. indica*, nerol (**20**) had high to very high anti-bacterial activity against *S. aureus*, *S. epidermidis*, *P. aeruginosa*, *B. subtilis*, *S. typhi*, and *K. pneumonia*, with MIC values ranging from 32 to over 64 µg/mL [11]. It is obvious that the action of some compounds in killing Gram-negative pathogens is more effective than that in Gram-positive pathogens. In short, α-elemene (**11**) inhibited the Gram-negative bacteria *S. enterica* and *E. coli* with a MIC value of 0.1 µg/mL and 25 µg/mL, respectively [6]. However, the Gram-positive bacteria, *S. typhimurium*, *B. cereus*, *S. aureus*, and *L. monocytogenes*, had a MIC value ranging from 60 µg/mL to 262.5 µg/mL [6,38]. The same phenomenon was also observed in *A. indica* and/or *A. dracunculus* for the low-very high anti-bacterial activities of carveol (**26**) [6,38], while α-farnesene (**13**), methyl linolenate (**72**), diisooctyl phthalate (**72**) had very low-very high anti-bacterial actions [6]. Camphene (**8**) [11,36] had low to high anti-bacterial activities.

Of note, 1,8-cineole (**28**) had low-high anti-bacterial activities against 37 bacterial pathogens [6,13,36,37] and this compound is present in *A. indica, A. feddei*, *A. vestica*, *A. iwayomogi*, etc [28]. Moreover, Alpha-pinene (**9**) had similar inhibition for Gram-negative and Gram-positive bacteria since its MIC had a low to very high activity. Gram-negative bacteria such as *K. pneumonia* (32 µg/mL), *P. aeruginosa* (64 µg/mL), *E. coli*^cs^ (98 µg/mL), *H. influenzae* (126 µg/mL), *S. typhi* (128 µg/mL), *S. dyssenteriae* (128 µg/mL), and *K. pneumoniae* (178 µg/mL)) and Gram-positive bacteria (*S. aureus* (32 µg/mL), *B. subtilis* (128 µg/mL), *S. epidermidis* (>128 µg/mL), *S. pyogenes* (132 µg/mL), MRSA^cs^ (172 µg/mL), *S. pneumoniae* (172 µg/mL), MRSA (210 µg/mL), MGRSA (256 µg/mL), and *S. aureus* (172 µg/mL)) [11,37]. As mentioned above, it also was similar condition for the anti-bacterial activity presented by β-pinene (**10**) and α-thujone (**29**) in *A. indica* which had a low to very high activity for Gram-negative and Gram-positive bacteria [11,17,37]. In addition, grandisol (**21**) and piperitone (**22**) had better results in *A. vestica* against 6 pathogens [37].

In parallel, dichloromethane extract of *A. integrifolia* was identified as the most active fraction. Six compounds were identified and their anti-bacterial activities for *E. coli*, *B. cereus*, *S. aureus*, and *Y. enterocolitica* are listed as follows in descending order: integracid (**73**) (MIC = 60–120 µg/mL) > zerumbone (**37**) (MIC = 90–230 µg/mL) > (+)-(*S*)-*ar*-turmerone (**35**) (MIC = 100–260 µg/mL) = α-curcumene (**14**) > (+)-(*S*)-dihydro-*ar*-turmerone (**36**) (MIC = 120–160 µg/mL) > dihydro-*ar*-curcumene (**15**) (MIC = 120–4000 µg/mL) [39]. Furthermore, a crude extract of *A. afra* inhibited Gram-positive bacteria (*A. naeslundii*, *A. israelii*, and *S. mutans*), and Gram-negative bacteria (*P. intermedia*, *P. gingivalis*, and *A. actinomycetemcomitans*) with a MIC value of 25 to 1600 µg/mL [9]. The highly active anti-bacterial action of the *A. afra* extract was attributable to six phytochemicals, including phytol (**46**), α-amyrin (**47**), and betulinic acid (**48**), acacetin (**50**), 12α,4α-dihydroxybishopsolicepolide (**39**), and scopoletin (**61**). All six compounds had very high anti-bacterial potential [9]. Especially, the three most active compounds, betulinic acid (**48**), acacetin (**50**), and scopoletin (**61**) which exerted an excellent antimicrobial activity with a MIC value of 0.3 to >1 µg/mL against the above mentioned pathogens [9]. Similarly, the sesquiterpenoid vulgarone B (**34**) found in *A. iwayomogi* species had a very high anti-bacterial activity against *S. typhimurium*, *S. aureus*, and *S. enteritidis* with a MIC value of 0.5 to >2 µg/mL [44].

The main compound in *A. herba-alba* oils, sesquiterpene germacrene D (**17**), was found against six bacteria, *B. subtillis*, *S. aureus*, *P. vulgaris*, *S. dyssenteriae*, *K. pneumonia*, and *S. typhi*, with MIC values varying from 30.31 to 90.15 µg/mL [32]. Mohamed et al. reported that six of seven secondary metabolites isolated from the organic extract of *A. herba-alba* aerial parts showed anti-bacterial activities. Using disc diffusion assays [46], two novel compounds, 1,3,8-trihydroxyeudesm-4-en-7α,11β*H*-12,6α-olide (**40**) and 3α,8β-dihydroxygermacr-4(15),9(10)-dien-7β,11α*H*,12,6α-olide (**41**) suppressed *B. subtilis* and *S. aureus* growth at 25 to 50 µg per disc and *B. subtilis*, *S. aureus*, and *E. coli* growth at 50 to 100 µg per disc, respectively. Similarly, the other compounds, 1β,8α-dihydroxy-11α,13-dihydrobalchanin (**42**), 11-epiartapshin (**43**), and 5-β-D-glucopyranosyloxy-7-methoxy-6*H*-benzopyran-2-one (**62**) could also inhibit the three pathogenic strains at 25 to 100 µg per disc, except that benzoic acid *p*-(β-D-glucopyranosyloxy)-methyl ester (**65**) at dose of 25 and 50 µg could kill *S. aureus* and *B. subtilis*, respectively [46]. In order to study the action of the *A. capillaris* extract on clinically drug-resistant bacteria, limonene (**6**&**7**), from this extract were shown to have very low to very high anti-bacterial activities against *E. coli*^cs^, *H. influenzae*, MRSA^cs^, *S. pyogenes*, *K. pneumoniae*, *S. pneumoniae*, MRSA, MGRSA, and *L. monocytogenes* with a MIC value of 20 to 332 µg/mL [34,35]. In addition, capillin (**64**), an aromatic ketone or ynone from *A. capillaris*, had moderate to high activities with a MIC value of 64 to 165 µg/mL inhibiting the listed pathogens [35].

Zhang et al. screened five fractions of aqueous extract of *A. argyi* leaf to find the most active fraction against *S. aureus*. As a result, its chloroform fraction showed a very low anti-bacterial activity with a MIC value of 3000 µg/mL [53]. One flavonoid, jaceosidin **(59)**, was detected among 24 phytochemicals from the chloroform fraction using LC-MS [53]. In addition, this compound was also found in *A. californica* and it had a MIC value of 3.3 µg/mL for *E. coli* Δ*tolC*, *E. coli* Δ*tolC* (*fabl*), and that of 33 µg/mL for *E. coli* WT [52]. Screening of these pathogens showed that jaceidin (**60**) from *A. californica* had very high anti-bacterial activity with a MIC value of 18 to 36 µg/mL [52]. However, chrysosplenol B (**54**) had very low to low anti-bacterial activity with a MIC value of 200 to 500 µg/mL [52]. More anti-bacterial compounds have been discovered in *A. argyi* species. Carveol (**26**), a monoterpenoid in the terpenoid group, had very high anti-bacterial activity towards *S. enterica*, *S. aureus*, *E. coli* ATCC25922, *S. typhimurium*, *E. coli* ATCC8739, *L. monocytogenes*, and *B. cereus* with a MIC value of 0.1, 15, 25, 30, 60, 100, and 120 µg/mL, respectively [6,38].

Two isomers β-ocimene (**3**&**4**) were major compounds of *A. dracunculus*; (*E*)-β-ocimene (**3**) had a lower MIC value of 130 to 650 for *B. subtillis*, *S. epidermidis*, *P. vulgaris*, *S. dyssenteriaem*, and *K. pneumonia* than (*Z*)-β-ocimene (**4**) with a MIC value of 130 to 600 [32]. Several other active compounds were purified from other *Artemisia* species. The *A. douglasiana* extract inhibited standard *H. pylori* and six clinical strains with a MIC value of 60 to 120 µg/mL. Consistently, its active phytochemical, dehydroleucodine had a MIC value of 1 to 8 µg/mL [45]. Moreover, five active compounds isolated from the *A. rupestris* extract, chrysosplenetin (**52**), chrysoeriol (**53**), penduletin (**56**), artemetin (**57**), and pachypodol (**58**) showed synergistic effects with norfloxacin, ciprofloxacin, and/or oxacillin against *S. aureus* isolates [51]. Two novel compounds ((*Z*)-2-(hexa-2,4-diyn-1-ylidene)-1,6-dioxaspiro(4.5)dec-3-ene (**68**) and (*E*)-2-(hexa-2,4-diyn-1-ylidene)-1,6-dioxaspiro(4.5)dec-3-ene (**69**)) were isolated from the acetone extract of *A. pallens* which had outstanding anti-bacterial activity against Gram-negative and Gram-positive infections such as *B. subtilis*, *S. aureus*, *P. aeruginosa*, and *E. coli* with a MIC value of 2.6–80.6 ug/mL, 4.7–10 µg/mL, 2.5–21.8 ug/mL, and 1.6–2.7 µg/mL, respectively [55]. In addition, thymol (**25**) was found to be one moderately active compound against *S. aureus* and *K. pneumoniae* K38^cs^ (MIC = 60 µg/mL), and *E. coli*, *P. aeruginosa*, and *A. baumannii* A52^cs^ (MIC = 80 µg/mL) in *A. haussknechtii* plant [42].

Together, based on the anti-bacterial activity, 64 out of 75 compounds can be categorized into 15 rankings although 11 compounds have so far not been tested for their MIC (Appendix A).

## 3. Mechanisms of Action of *Artemisia* Plants and Their Compounds

*Artemisia* plants and their phytochemicals are well known for their antimicrobial properties [9,28,39,46]. More and more studies report that *Artemisia* plants and their phytochemicals possess a variety of anti-bacterial mechanisms [37,52,55,59,65]. They include destruction of cell wall, membrane, and cytosol, morphological changes, decreased virulence, and interference with DNA, protein, and cell division in bacteria as shown in Figure 2.

### 3.1. Targeting Cell Membrane/Cell Wall

The hydrophobicity of solutions must pass a cell membrane and the membrane’s structure dictates a cell membrane’s permeability [66]. A large volume of literature demonstrates that the *Artemisia* essential oils are able to damage the cytoplasmic membrane in bacteria and cause their loss of vital chemicals and cell death [67]. The anti-bacterial action of the essential oils can be ascribed to multiple mechanisms since there are different potential targets in bacteria. As described in Section 1.2. “Chemical compositions”, *Artemisia* plants are rich in terpenes with potent anti-bacterial properties. Terpenes and terpenoids are partitioned into the layer of essential oils. Terpenes and terpenoids have the capability to increase membrane permeability by inserting through the phospholipidic bilayer in bacteria and, consequently, the leaking cell membrane causes the loss of cellular contents and cell lethality in bacteria [40]. For example, limonene (**6**&**7**) inhibited *L. monocytogenes* and other food-borne pathogens due to the interruption of bacterial cell integrity and wall structure observed using scanning electron microscopy (SEM) [34]. Limonene (**6**&**7**) enhanced the conductivity and induced nucleic acid and protein leakage that results in damaging cell membrane permeability and cell membrane rupture based on conductivity and PI staining measurements [34]. Furthermore, (+)-limonene (**6**) could lower inner pH values. pH 4.0 is more effective in eliminating *E. coli* BJ4. The *E. coli* MC4100 lptD4213 mutant with enhanced outer membrane permeability showed higher sensitivity to (+)-limonene (**6**) at pH 4.0. Furthermore, reflectance infrared microspectroscopy revealed that β-sheet proteins played a crucial part in the mechanism of (+)-limonene (**6**). *E. coli* BJ4′s resistance to (+)-limonene was not altered by rpoS deletion, sub-lethal heat, acid shock, nor both of these conditions [68].

In order to inactivate *E. coli* O157:H7 in fruit juices or preserved foods [68], a synergistic combination procedure with heat was created based on the investigation of the mechanism of inactivation by (+)-limonene (**6**). Active compounds such as α-elemene (**11**), carveol (**26**), α-farnesene (**13**), methyl linolenate (**72**), diisooctyl phthalate (**66**), camphene (**8**), and 1,8-cineole (**28**) showed lower anti-bacterial activities on Gram-negative pathogens than Gram-positive cells. This may be due to the fact that the Gram-positive bacteria have a thicker layer of peptidoglycan that makes compounds more difficult to pass through and leads to imparting rigidity to the bacteria [69]. Yang et al. reported that methanolic crude extract of *A. indica* and its active major compounds, including carveol (**26**), 1,8-cineole (**28**), α-elemene (**11**), methyl linolenate (**72**), α-farnesene (**13**), and diisooctyl phthalate (**66**) could damage or kill *E. coli*, *S. enterica*, and *L. monocytogenes* as evidenced via membrane destruction based on the PI staining method [6]. Their transmission electron microscope (TEM) data showed that the *A. indica* crude extract caused bacterial emptiness while its most active compound, carveol (**26**) also led to severe bacterial membrane defects including membrane poring, wrinkling, emptiness, and membrane discontinuities [6]. Previous study showed that there were also changes in cell morphological under treatment with carveol (**26**) in *S. aureus*, *E. coli*, and *S. typhimurium* as shown by SEM with irregularly sized cells and the presence of debris in *E. coli* suggesting that cell division disruption or cellular membrane malfunction may have occurred [40].

Hydoxyl groups are highly reactive in some terpenoids (monoterpenoids) such as thymol (**25**), carveol (**26**), terpineol (**24**); as well as eugenol (**75**) in the other group (Table 4). The hydrogen bonds indicates for the active sites which targeting the enzymes, causing protein inactivation and cell membrane rupture or malfunction in bacteria [40,70,71]. For instance, a strong anti-bacterial compound, thymol (**25**) (Appendix A) could inhibit Gram-negative and Gram-positive bacteria, including *E. coli*, *A. baumannii* A52^cs^, *S. aureus*, *P. aeruginosa*, and *K. pneumoniae* K38^cs^ [42]. The SEM data demonstrated that the mode of action of thymol (**25**) with the hydroxyl group at a distinct position on the phenolic ring involved the membrane dysfunction in *S. typhimurium* [72] and disturbance of membrane integrity and impaired permeability that lead to leakage of membrane potential, protons, K^+^ ions, and ATP in *E. coli* [73]. Eugenol (**75**) and terpineol (**24**) also caused cell death through disrupting cellular membrane and function as the cell membrane was entirely destroyed and surrounded by cell debris in *S. typhimurium* [40]. Although a sesquiterpenoid, vulgarone B (**34**), did not cause leakage of a significant cytoplasmic component, the SEM data indicated that it indeed altered cell morphology of *S. aureus* which became bloated and, crushed and aggregated at 8 h and 24 h post treatment, respectively. This led to 99.62% cell death in *S. aureus* indicating that vulgarone B (**34**) had strong bacterial effects on *S. aureus* without significantly breaking the cell membrane [44].

RA (**74**), a compound that grouped as miscellaneous showed high MIC values against *S. aureus, E. coli,* and MRSA as aforementioned, which might be explained by its inefficient penetration capability into bacterial cell walls. Interestingly, RA was found to have synergy, particularly in the log phase of bacterial growth, with antibiotics including amoxicillin, vancomycin, and ofloxacin against *S. aureus* and exclusively with vancomycin against MRSA [59]. In a different study, this acid induced membrane damage, resulting in impaired cell wall and membrane in *S. aureus* and leakage of bacterial contents and ions [74]. Antimicrobial activity of phenolic compounds was mainly attributed to inactivating cellular enzymes, which depends on the speed of penetration into the cell or its ability to alter membrane permeability [75].

### 3.2. Targeting DNA

DNA agarose gel analysis revealed that 8 to 24 h incubation of *S. aureus* with vulgarone B at 1000 µg/mL induced DNA breakage [44]. In addition, vulgarone B (**34**) was found to cause a single DNA nick and change in DNA mobility shift, indicating a mutual interaction or DNA breakage [44]. While the (−)-limonene (**7**) has been shown to be more active than (+)-limonene (**6**). The (−)-limonene (**7**) damaged DNA, which induced SOS response, membrane impairment and release of heat shock proteins (HSPs) as evidenced by the induction of PkatG and PsoxS promoters in *E. coli* models via formation of reactive oxygen species. At high concentrations, (−)-limonene (**7**) causes irreversible degrading processes in both *S. aureus* (Gram-positive) and *E. coli* (Gram-negative) for 24 h. This phenomenon was observed to be weaker in *E. coli* when treated with α-pinene (**9**) compared to inducing only heat shock [76]. Thymol (**25**) repressed the *hilA* gene, which encodes a gene activator for the virulence of *S. typhimurium*, increased DNA thermal stability, and inhibited transcription by downregulating the DNA-binding protein H-NS [73].

### 3.3. Target Protein and Enzymes

Alpha-pinene (**9**) at 2.72 µg/mL only partially inhibited protein refolding in *E. coli* with a HSP IbpB mutation. However, (−)-limonene (**7**) (1.36 µg/mL) competed with IbpB to bind to hydrophobic sites of DnaKJE chaperone and, thus, inhibited the DnaKJE –ClpB bichaperone-dependent refolding function of heat-inactivated bacterial luciferase in *E. coli* WT and mutant ∆ibpB strains. Furthermore, (−)-limonene (**7**) (0.136 µg/mL) induced the overproduction of hydrogen peroxide and superoxide anion radicals, eventually, leading to DNA and protein damage in *E. coli* as detected by inducible specific lux-biosensors. The induction of oxidative stress in the first minute of (−)-limonene (**7**) could be related to the synthesis of reactive oxygen species (ROS), which has been reported in several antibiotic cases. Moreover, a significant increase in the inhibitory effect of (−)-limonene (**7**) was observed without catalase and peroxidase enzymes in bacterial strains JW3914-1 ΔkatG729::kan, especially JW3933-3 ΔoxyR749::kan, indicating that hydrogen peroxide played an important role. In addition, both terpenes could induce heat shock to damage the cells [76]. Limonene (**6**&**7**) declined the activity of electron transfer chain (ETC), composed of complexes I to V, located on the plasma membrane of *L. monocytogene* since it significantly downregulated the protein level of complexes III, IV, and V, and some protein units in complexes I and II following 24-h treatment using ESI MS/MS [34]. Limonene inhibited the activity of the ETC complex and ATPase in *L. monocytogenes*, resulting in a decrease in ATP content and intracellular ATPase activity (Na^+^K^+^-ATPase, Ca^2+^-ATPase) [34]. Limonene might inhibit respiration by blocking the electron transmission from NADH to coenzyme Q, which might explain why ATP synthesis is blocked by limonene (**6**&**7**).

The increased permeability of the cell membrane was shown to delay the capacity to produce essential compounds for growth and reproduction and, eventually, cell death in bacteria. In light of the aforementioned findings, limonene might decrease enzyme activity, limit respiration, and mess with the ATP balance in *L. monocytogenes* [34]. Particularly, the expression of the complex I subunit (Unigene11357 CK 0A, CL1094.Contig4 CK 0A, CL1528.Contig4 CK 0A, and CL4703.Contig1 CK 0A), in charge of acquiring two electrons from NADH and transferring them to coenzyme Q via ferritin, was markedly increased, indicating that more electrons from NADH would be transported into the ETC of the *L. monocytogenes* [34] by limonene (**6**&**7**). Accordingly, limonene (**6**&**7**) blocked the ETC and accumulated electrons in the cytochrome (CL594.Contig2 CK 0A), one subunit of the complex III, and the cytochrome oxidase subunit (Unigene2340 CK 0A, Unigene7527 CK 0A, CL3277.Contig1 CK 0A) of the complex IV. A considerable downregulation of the majority of ATP synthase subunits in complex V further implied that ATP synthesis was inhibited, which was in good agreement with the drop in ATP content. Such treatment also caused considerable downregulation of the Unigene6313 CK 0A subunit of complex V’s V-type proton ATPase. The V-type proton ATPase hydrolyzed ATP to produce an electrochemical gradient across the membrane in addition to controlling pH within and outside of bacteria [34]. Normal bacteria experienced necrosis and apoptosis when exposed to the high levels of extracellular H^+^ produced by the milieu created by a high level of V-ATPase [77].

Through hydrophilic and hydrophobic interactions, thymol (**25**) could interact with membrane bound or periplasmic proteins [78]. For example, thymol (**25**) upregulated the levels of OmpA, OmpX, GlnH, and FabI that are related to the synthesis of outer membrane proteins [73]. The build-up of misfolded outer membrane proteins as well as the increase of gene expression in outer membrane protein production was observed in *S. enterica* after exposing it to thymol (**25**) at sub-lethal concentrations. Moreover, thymol (**25**) affected the citrate metabolic pathway, which eventually affected ATP synthesis. Thymol (**25**) has been shown to alter various pathways of cell metabolism and impair the metabolic pathway. For example, it downregulated the protein PtsH involved in the phosphotransferase system and the sugar transport system, upregulated the proteins Enolase (Eno) and 2,3-bisphosphoglycerate-independent phosphoglyceromutase (iPGM), and downregulated the ATP synthase α-subunit, 2,3-bisphosphoglycerate-dependent phosphoglycerate mutase dPGM and glyceraldehyde-3-phosphate dehydrogenase A (AtpA, GpmA and GapA) and the dPGM involved in energy metabolism.

The citrate breakdown route involved the enzymes S-adenosylmethionine synthetase and the autonomus glyacyl radical cofactors (MetK and GrcA). In addition, they proposed that downregulation of S-adenosylmethionine synthase. The autonomus glyacyl radical cofactors (MetK and GrcA) implicated in the citrate degradation pathway would suppress the activation of the pyruvate formate lyase, resulting in a blockage of the pathway and a downregulation of AckA, and that a build-up of citrate would result in CitE overexpression. Since acetate kinase was involved in the production of ATP, it is possible that thymol plays a significant role in the dysfunction of this metabolic process [73]. Thymol (**25**) inhibited protein biosynthesis by upregulating the 30S ribosomal protein S1 (RpsA), which was involved in translation initiation and elongation processes, and down-regulating the 50S ribosomal protein L7/L12 (RplL), the binding site for several factors involved in protein synthesis and translation accuracy. According to these results, thymol (**25**) might regulate cell wall synthesis that might be linked to cell division with central metabolism.

RA (**74**) was shown to completely suppress the expression of surface membrane proteins (MSCRAMM’s), 40 to 90 kDa, main factors that cause infections. They are covalently anchored to RA, making them prime targets for antibiotics since they mediate the initial host-bacterial interactions and resistance in MRSA leading to suppression of virulence factors [59]. Other compounds, chrysosplenetin (**52**), chrysoeriol (**53**), and penduletin (**56**) were found to bind protein NorA, a cytoplasmic membrane containing a multidrug efflux protein and reassemble proteoliposomes [79]. This binding could change the morphology and function of NorA [51]. To develop novel antibiotics that generally target fatty acid biosynthesis and FAS II enzyme enoyl reductase in bacteria is thought to be a promising strategy [80,81]. Thus, jaceosidin (**59**) was examined and found to inhibit FabI in vitro, but no signal was found in vivo against the enzyme in *E. coli* [52]. Both spiro compounds (*Z*)-2-(Hexa-2,4-diyn-1-ylidene)-1,6-dioxaspiro(4.5)dec-3-ene (**68**), and (*E*)-2-(Hexa-2,4-diyn-1-ylidene)-1,6-dioxaspiro(4.5)dec-3-ene (**69**) inactivated Gram-positive (*B. subtilis* and *S. aureus*) and Gram-negative (*P. aeruginosa* and *E. coli*) bacteria, perhaps since these compounds could bind to the active site of the DNA gyrase B, an important bacterial enzyme that catalyzed the negative supercoiling of double-stranded closed-circular DNA in an ATP-dependent manner [82] using molecular docking [55].

### 3.4. Others

There were also several case studies of *Artemisia* species acting as inhibitors of both bacterial growth and biofilm formation. For instance, 24 h treatment with essential oils of *A. herba-alba, A. campestris,* and *A. absinthium* which major component were 32.07% β-pinene (**10**), 39.21% chamazulene, and 29.39% α-thujone (**29**), at 620 µg/mL could reduce biofilm formation by up to 45% for *E. coli* E2346/69 and 70% for *E. coli* K-12 [24]. In particular, eugenol (**75**) at 200 µg/mL destroyed the cell membrane of carbapenem-resistant *Klebsiella pneumoniae* (CRKP), as characterized by decreased intracellular ATP concentration, reduced intracellular pH and cell membrane hyperpolarization, coupled with enhanced membrane permeability, and led to changes in bacterial cell structure and intracellular component leakage. Additionally, eugenol (**75**) inhibited biofilm formation and inactivated biofilm in CRKP by using different EM techniques. This happened due to the fact that eugenol (**75**) strongly inhibited biofilm-associated gene expression (*pgaA*, *luxS*, *wbbM*, and *wzm* genes of CRKP-12), upregulated *mrkA* mRNA, and inactivated CRKP cells growing in biofilms [60]. In the case of thymol (**25**), it was also reported to inhibit biofilm formation against different carbapenemase-producing Gram-negative bacilli [78,83,84].

Estragole (**73**) has been proposed to be a possible inhibitor of the efflux pump against *S. aureus* 1199B and *S. aureus* K2068 species. This inhibition is supported by data indicating that estragole (**73**) treatment enhanced the action of norfloxacin and ethidium bromide against *S. aureus* 1199B and lowered the MIC of ethidium bromide against *S. aureus* K2068 [85]. Artesunate (**45**) has been shown to increase the antibiotic properties of β-lactams against MRSA by binding penicillin-binding protein 2a (PBP2a) and downregulating MecA expression, which was upregulated by oxacillin. In addition to reducing bacterial load, artesunate (**45**) was found to protect mice from MRSA WHO-2 (WHO-2) when combined with oxacillin. In mouse peritoneal macrophages stimulated with heat-killed WHO-2, artesunate (**45**) inhibited the expression level of TLR2 and Nod2, two key players in the inflammasome, suggesting artesunate (**45**) is a candidate drug for MRSA sepsis [86]. The data obtained from confocal microscopy and liquid chromatography-tandem mass spectrometry (LC-MS/MS) revealed that artesunate (**45**) increased the accumulation of antibiotics (daunorubicin and oxacillin) in MRSA, suggesting that artesunate (**45**) could affect the efflux pumps of antibiotics. Consistently, artesunate (**45**) inhibited the gene expression of NorA, NorB, and NorC but not MepA, SepA, and MdeA, in the efflux pumps [48]. A working model of artesunate is delineated in Figure 3.

The limitations and challenges faced in the development of new anti-bacterial compounds from *Artemisia* plants include: (1) *Artemisia* plant-derived phytocompounds generally have low anti-bacterial potency; (2) anti-bacterial mechanisms of action of the compounds are not clear; (3) anti-bacterial compounds with high potency need more labor and time to be identified; and (4) total synthesis of the anti-bacterial compounds such as terpenoids is challenging [87,88]. Additionally, *Artemisia* plants have been used to treat bacterial infections in humans from ancient time. So far, only artemisinin from *A. annua* has been developed as a prescription drug against malaria. Some gaps in development of anti-bacterial drugs from the *Artemisia* plants include the identification, safety, efficacy, and synthesis of active compounds from this genus. Currently, components of the essential oils form this genus was successfully identified and they had a MIC of 0.1 μg/mL against certain bacteria but not the other. This suggests other potential anti-bacterial compounds need to be characterized, which may reflect the gaps from current findings to drug discovery [89,90].

## 4. Conclusions

The genus *Artemisia* comprises over 500 species. These plants are annual or perennial aromatic herbs and subshrubs with greenish to yellowish leaves, white or yellow flowers, and small black seeds. *Artemisia* species are a remarkable source of foods and medicines and their culinary and medicinal functions can be attributed to their rich phytochemicals. Despite significant advances in phytochemical and biological research of *Artemisia* plants over recent years, comprehensive and critical reviews of this genus are fragmented or limited. The present review updated and summarized information about the chemistry, anti-bacterial properties, and mechanisms of action of the *Artemisia* plants and phytocompounds. Seventy-five compounds present in twenty *Artemisia* species were extensively dis-cussed with regard their chemical structure, anti-bacterial activity and mechanism and structure-and-activity relationship. Generally speaking, compounds of the *Artemisia* plants inhibit bacteria more potently than their crude extracts. However, the structure of these compounds also affects this bacterial inhibition as well as their modes of action. Caution should be taken in the use of the *Artemisia* plants and phytochemicals for bacterial infections in humans and animals.

## Figures and Tables

**Figure 1 bioengineering-10-00633-f001:**
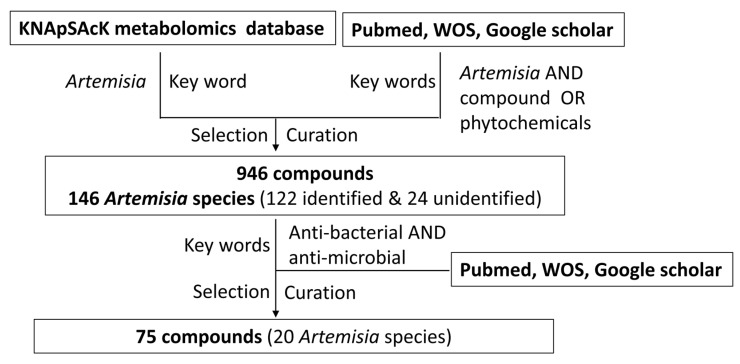
Schema delineating the characterization of anti-bacterial compounds from the *Artemisia* species. In the first step, we selected and curated the phytochemicals of the *Artemisia* plants from the KNApSAcK metabolomics databases in cross-reference with other databases (Pubmed, Google scholar and Web of Science (WOS)). In the second step, we checked 946 compounds with text search into Pubmed, Google scholar, and Web of Science (WOS) using anti-bacterial or antimicrobial key words. As a result, 75 compounds from this genus were identified and classified based on their chemical structures.

**Figure 2 bioengineering-10-00633-f002:**
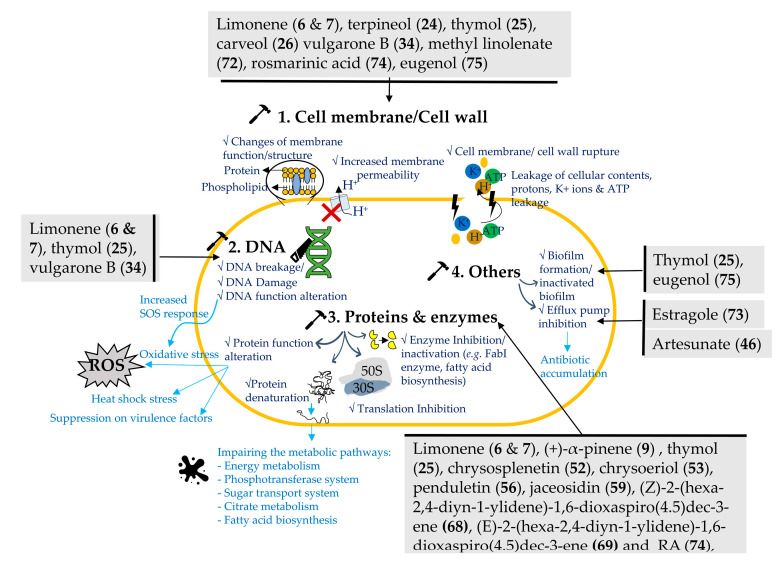
Anti-bacterial mechanisms of the *Artemisia* compounds. Four likely mechanisms by which the *Artemisia* phytochemicals inhibit pathogenic bacteria are proposed in different molecular targets and compartments of bacteria. (1) *Artemisia* essential oils and 9 constituents target bacterial cell membrane and/or cell wall via changes in membrane structure and function, membrane permeability and rupture of bacterial membrane and/or cell wall, eventually leading to cause the release of vital chemicals and cell death; (2) (+)-Limonene (**6**&**7**), vulgarone B (**34**) and thymol (**25**) can interfere with DNA structure and function, and, in turn, induce SOS response, and oxidative stress (i.e., over-production of reactive oxygen species (ROS)). Consequently, aberrant ROS kills bacteria; Limonene (**6**&**7**), (+)-α-pinene (**9**), thymol (**25**), chrysosplenetin (**52**), chrysoeriol (**53**), penduletin (**56**), RA (**74**)**,** jaceosidin (**59**), (*Z*)-2-(Hexa-2,4-diyn-1-ylidene)-1,6-dioxaspiro(4.5)dec-3-ene (**68**), and (*E*)-2-(Hexa-2,4-diyn-1-ylidene)-1,6-dioxaspiro(4.5)dec-3-ene **(69)** can target proteins and enzymes via altering protein functions, denaturing proteins and reducing enzyme activities. Consequently, oxidative stress, heat shock stress, and expressional inhibition of virulent factors are induced in bacteria; and (4) four compounds target the other mechanisms. Thymol (**25**) and eugenol (**75**) can target biofilm pathways via reduction and inactivation of biofilm, leading to loss of bacterial virulence and survival. Estragole (**73**) and artesunate (**45**) inhibit the efflux pumps and, as a result, accumulate antibiotics within bacteria.

**Figure 3 bioengineering-10-00633-f003:**
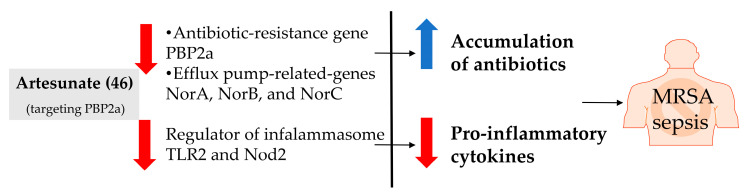
A schema illustrating the mechanisms by which artesunate protects mice against MRSA and its sepsis in mice. Artesunate (**45**) could enhance the antibiotic activity of β-lactams against MRSA via binding PBP2a. Two likely pathways of artesunate (**45**) are proposed below. (1) Artesunate (**45**) downregulates the expression level of NorA, NorB, NorC, and PBP2a, impairs the antibiotic efflux and enhances antibiotic activities against drug-resistant bacteria such as MRSA. (2) Artesunate (**45**) decreases the expression level of TLR2 and Nod2, two crucial players in the inflammasome, and, thus, diminishes inflammation. Consequently, artesunate can treat MRSA infection and, in turn, MRSA-induced sepsis in hosts (adopted from the publication [48]).

**Table 1 bioengineering-10-00633-t001:** Taxonomy of *Artemisia* plants.

Kingdom	Plantae
Division	Magnoliophyta
Class	Magnoliopsida
Order	Asterales
Family	Asteraceae
Genus	*Artemisia*
Species	>500 species

## Data Availability

The data supporting the conclusions of this review are included in the literature.

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
