# Peer review of "Phytochemistry, Pharmacology and Mode of Action of the Anti-Bacterial Artemisia Plants"

_bioengineering, 2023, doi:10.3390/bioengineering10060633_

Round 1

Reviewer 1 Report

In this review article, the authors summarized recent information about the phytochemistry, pharmacology, and toxicology of the Artemisia plants from 2003 to 2022. Twenty Artemisia species and 75 compounds have been documented to possess anti-bacterial functions and multiple modes of action. The compounds were classified into terpenes, polyphenols, and so on. The Mechanisms of action of Artemisia plants and their compounds were presented in a beautiful figure. Overall, this review was well-written and could be accepted after revision. Here are the suggestions:

1.       There are some language mistakes. P4, Line 160, delete “of” after contain.

2.       Table 2, The MW of compound 7 is missing.

3.       Table 2, the structure of compound 48 is not well presented.

4.       Table 3, the structure of compound 62 is not well presented.

5.       Table 4, the structure of compound 68 is not well presented.

6.     Can authors change the figures to high-resolution ones?

Author Response

Dear Reviewer,

We thank you for your precious time and efforts to help us improve our manuscript. First of all, we provided the point-to-point reply to the reviewers’ comments. Please note that all the modifications are indicated based on the lines and pages (or the serial number (SN) for the table) of the clean version of revised manuscript (Artemisia-2402571-clean). Besides, the tracked version of the revised manuscript (Artemisia-2402571-Track-change) and the PTP_reply is provided as additional files for your reference. Thanks again for your great help.

Reviewer 1.

Point 1: There are some language mistakes. P4, Line 160, delete “of” after contain.

Response 1: “of” was deleted (line 154, page 4).

Point 2: Table 2, The MW of compound 7 is missing.

Response 2: The MW of Compound 7 (136.2) was added to Table 2 (page 5).

Point 3: Table 2, the structure of compound 48 is not well presented.

Response 3: We have corrected the structure of Compound 48 in Table 2 (page 14).

Point 4: Table 3, the structure of compound 62 is not well presented.

 Response 4: We have corrected the structure of Compound 62 in Table 3 (page 16).

Point 5: Table 4, the structure of compound 68 is not well presented.

Response 5: We have corrected the structure of Compound 68 in Table 4 (page 17).

Point 6: Can authors change the figures to high-resolution ones?

Response 6: All the Figures (fig. 1, page 4; fig. 2 page 22; fig. 3 page 26) were changed into high resolution with 1000 dpi.

Reviewer 2 Report

The article is generally well written and suitable for publication in the journal "Bioengineering". I just have a single but significant query. The authors mention that this article is focused on mechanistic studies to understand the chemical nature of these compounds. but these approaches are missing from the article. It is acceptable if authors provide a little more attention to this.

Author Response

Dear Reviewer,

We thank you for your precious time and efforts to help us improve our manuscript. First of all, we provided the point-to-point reply to the reviewers’ comments. Please note that all the modifications are indicated based on the lines and pages (or the serial number (SN) for the table) of the clean version of revised manuscript (Artemisia-2402571-clean). Besides, the tracked version of the revised manuscript (Artemisia-2402571-Track-change) and the PTP_reply is provided as additional files for your reference. Thanks again for your great help.

Reviewer 2

Point 1: The article is generally well written and suitable for publication in the journal "Bioengineering". I just have a single but significant query. The authors mention that this article is focused on mechanistic studies to understand the chemical nature of these compounds. but these approaches are missing from the article. It is acceptable if authors provide a little more attention to this.

 Response 1: Thanks to Reviewer for this advice. The mechanism of the compounds is important and included in this manuscript. We included as many mechanistic studies of the compounds as possible. However, most of the compounds lack the modes of action in the literature and thus we failed to present their modes of action. Instead, we presented a picture of their mechanisms in Figures 2 (line 356, page 22) and 3 (line 578, page 26).

Reviewer 3 Report

Dear Authors, please check the following comments.

1-In general, the authors should be very careful with the writing style and revise the chemical names and write them correctly in the whole MS, also, the stereochemistry of the given structures needs to be revised. the MICs values for the total oils rather than individual are very important and should be mentioned in the discussion.

2- Line 37, please correct AMR (from where M comes from?)

3- lines 60-61 "reported to have been used to control fevers" (correct).

4- line 90- correct

5- line 94 change "properties" to composition

6- lines 94-97 please re-phrase. the authors should be careful "not all listed class of compounds are aromatic or volatile"

7- lines 102/108, Z is italic; please check the whole MS (also as P (should be small case), H (italics) ..etc

8- line 107, p-allylanisole, p is italic, and E is italic in ((E)-β ocimene), and other places.

9- lines 108-110 rephrase, 

10- line 117; α-amyrin (47), betulinic acid (48), acacetin (56), are not volatile compounds and are not belong to essential oils.

11- line 126; remove "in the essential oils"

12- line 125; thujone should be thujones

13- line 141/153/154; what is the difference between "terpenes and terpenoids" please correct and change accordingly. Please try to get accurate and correct references to support the manuscript}. some references tried to discriminate between the two terms but were not correct (they are not two classes of compounds).

14- Paragraph 1.3.1 is not accurate and needs careful revision.

15- table 2. MIC (μg/ml); change l o L. The names and stereochemistry need careful revision. 

16- check the names/structures of 40-45; 47-49.

17- The activities of compounds 52, 53, 56-58, are not exist, please check or remove them.

18- the name of compound 65 is incomplete.

19- paragraph 192-201, please add the values of the MICs

20- the paragraph (lines 250-280) is very congested, and needs careful revision and rephrasing.

21- line 286, remove "a carboxylic acid", and also, remove all the identified classes of compounds before the name of the compound, for example, lines 293-295 "a diterpenoid: phytol (46), two triterpenoids :  α-amyrin (47), and betulinic acid (48), and a flavonoids : acacetin (50), a sesquiterpenes : 12α,4α-dihydroxybishopsolicepolide (39), and a coumarin : scopoletin (61)." remove the highlighted words and all the text.

22- line 613-authors added caution to using Artemisia species for humans. the authors may need to explain more, and if any of the species reported to be toxic to human

Author Response

Dear Reviewer,

We thank you for your precious time and efforts to help us improve our manuscript. First of all, we provided the point-to-point reply to the reviewers’ comments. Please note that all the modifications are indicated based on the lines and pages (or the serial number (SN) for the table) of the clean version of revised manuscript (Artemisia-2402571-clean). Besides, the tracked version of the revised manuscript (Artemisia-2402571-Track-change) and the PTP_reply is provided as additional files for your reference. Thanks again for your great help.

Reviewer 3

Point 1: In general, the authors should be very careful with the writing style and revise the chemical names and write them correctly in the whole MS, also, the stereochemistry of the given structures needs to be revised. the MICs values for the total oils rather than individual are very important and should be mentioned in the discussion.

 Response 1: Thanks reviewer for your advice. The chemical names and structures have corrected (Table 2, SN 3,4,5 page 5; SN 15, page 7; SN 27, 28, page 9; SN 30 page 10; SN 35-36, 40, page 12; SN 41, 43, page 13); (Table 3, SN 62 page 16); (Table 4, SN 65, 68, 69, page 17, SN  71, page 18). The MICs values of essential oils (i.e., total oils) of some Artemisia plants were indicated in the discussion part (Lines 203-204 page 19; lines 246-247, page 19; lines 547-549, page 26). Of note, the MICs values of essential oils of the other Artemisia plants were not reported.

Point 2: Line 37, please correct AMR (from where M comes from?)

Response 2: AMR is an abbreviation of antimicrobial resistance. We changed antibiotics into antimicrobial resistance (line 37, page 1).

Point 3: lines 60-61 "reported to have been used to control fevers" (correct)

Response 3: We have corrected this sentence into “reported to be used to control fevers (lines 59-60, page 2).

Point 4: line 90- correct

 Response 4: We have corrected the sentence as follows “To date, around 1,340 plants including the Artemisia plants have been claimed to possess anti-bacterial activities (line 87-88, page 2)

Point 5: line 94 change "properties" to composition.

Response 5: We have changed “properties” to “composition” (line 91, page 3).

Point 6: lines 94-97 please re-phrase. the authors should be careful "not all listed class of compounds are aromatic or volatile"

Response 6: We have re-phrased the sentences as follows. “As most Artemisia species are aromatic, they are rich in aromatic and volatile compounds though non-aromatic and non-volatile compounds exist. Basically, most of the identified compounds consist primarily of terpenoids, flavonoids, coumarins and others (miscellaneous group). For instance, the major compounds in the A. annua essential oils were monoterpenoids including 30.7% artemisia ketone (18), 15.8% camphor (33), and 18.2% sesqueiterpenes [23] (line 91-96, page 3).

Point 7: lines 102/108, Z is italic; please check the whole MS (also as P (should be small case), H (italics), etc

Response 7: Per Reviewer’s advices, we have italicized E (lines 103, page 3; Table 2, SN 3 page 5; Table 4, SN 69, page 17; lines 327, 337 page 21; line 367, page 22; line 539 page 25), Z (lines 98 page 3; lines 104, 111, page 3; Table 2, SN 4, page 5; Table 2, SN 27, page 9; Table 4, SN 68 page 17; line 248 page 19; lines 329, 336 page 21; line 366 page 22; line 538 page 25) and put P into an italicized small letter (lines 103 page 3; Table 2, SN 5, page 5; Table 4, SN 65 page 17; line 250 page 19; line 306 page 21) and italicized H (Table 2, SN 40, 41 page 13; Table 3, SN 62 page 16; lines 301, 305 page 20) throughout the revised manuscript.

Point 8: line 107, p-allylanisole, p is italic, and E is italic in ((E)-β ocimene), and other places.

Response 8: We have corrected them as Reviewer’s advices as the response to Question 7

Point 9: lines 108-110 rephrase

Response 9: We have re-phrased the sentences below. Similarly, the gas chromatography (GC) and mass spectroscopy (MS) analysis [25] indicated that the main essential oils of A. argyi included 16.2% 1,8-cineole (28), 14.3% β-pinene (10), 14% camphor (33), 13.9% artemisia ketone (18), and 11.1% α-pinene (9) (lines 104-107, page 3).

Point 10: line 117; α-amyrin (47), betulinic acid (48), acacetin (56), are not volatile compounds and are not belong to essential oils.

Response 10:  We corrected this mis-concept and have modified the sentences as follows (lines 112-115, page 3). Therefore, Artemisia plants are rich in lipid-soluble components, especially in their essential oils. In addition, phytol (46), α-amyrin (47), betulinic acid (48), acacetin (56), 12α,4α-dihydroxybishopsolicepolide (39), and scopoletin (61) were isolated from A. afra [9].

Point 11: line 126; remove "in the essential oils"

Response 11: “in the essential oils” was removed (line 121, page 3).

Point 12: line 125; thujone should be thujones

Response 12: We have changed “thujone” became “thujones” (line 121, page 3).

Point 13: line 141/153/154; what is the difference between "terpenes and terpenoids" please correct and change accordingly. Please try to get accurate and correct references to support the manuscript. some references tried to discriminate between the two terms but were not correct (they are not two classes of compounds).

Response 13:  Thanks to reviewer for the suggestion. Here we define the different between terpenes and terpenoid and re-phrased the sentence according to the references. “Terpenes, a simple hydrocarbons structures, while terpenoids (oxygen-containing hy-drocarbons) are defined as modified class of terpenes with various functional groups and oxidized methyl groups moved or deleted at various places which classified into alcohols, ethers, aldehydes, phenols ketones, esters, and epoxides that are volatile [30,31].” (Lines 150-154, page 4)

Point 14: Paragraph 1.3.1 is not accurate and needs careful revision.

Response 14:  We have changed the sentences with “Forty-nine terpenes and terpenoids with anti-bacterial properties found in Artemisia plants are listed in Table 2. They constitute the majority of compounds in Artemisia as described in Section 1.2 (Chemical composition). Terpenes, a simple hydrocarbons structures, while terpenoids (oxygen-containing hydrocarbons) are defined as modified class of terpenes with various functional groups and oxidized methyl groups moved or deleted at various places which classified into alcohols, ethers, aldehydes, phenols ketones, esters, and epoxides that are volatile [30, 31]. Terpenes contain ten monoterpenes and seven sesquiterpenes. There are also compounds identified as terpenoids, consisting of sixteen monoterpenoids, twelve sesquiterpenoids, one diterpenoids and three triterpenoids as listed in Table 2.” (lines 148-157, page 4).

Point 15: table 2. MIC (μg/ml); change l to L. The names and stereochemistry need careful revision.

Response 15: We corrected the μg/ml into μg/mL throughout the revised manuscript (Table 2, page 4; Table 3 page 15; Table 4 page 16) and in the whole MS. The compound name also was revised (Table 2, SN 3,4,5 page 5; SN 15, page 7; SN 27, 28, page 9; SN 30 page 10; SN 35-36, 40, page 12; SN 41, 43, page 13); (Table 3, SN 62 page 16); (Table 4, SN 65, 68, 69, page 17, SN  71, page 18).

Point 16: check the names/structures of 40-45; 47-49.

Response 16: We have corrected their names and structures in Table 2 (SN 40-45, pages 12-13; SN 47-49, page 14)

Point 17: The activities of compounds 52, 53, 56-58, are not exist, please check or remove them.

Response 17: The MIC values of compounds 52, 53, and 56-58 were not reported in the reference (Lan et al., (2019)). However, the MIC values of these compounds in combination with other antibiotics against five drug-resistant S. aureus strains were reported (lines 333-336, page 21). Due to the importance of the above compounds, we decided to keep them in Table 3.

Point 18: the name of compound 65 is incomplete.

Response 18: We have corrected the name of Compound 65 “Benzoic acid p-(β-D-glucopyranosyloxy)-methyl ester” (Table 4, SN 65, page 17).

Point 19: paragraph 192-201, please add the values of the MICs

Response 19: The MICs values from the crude extract of the A. annua methanolic and ethanolic extracts were not reported and could not be added back (line 185-195, page 18).

Point 20: the paragraph (lines 250-280) is very congested, and needs careful revision and rephrasing.

Response 20: We have re-phrased the paragraph (lines 244-276, pages 19-20).

Point 21: line 286, remove "a carboxylic acid", and also, remove all the identified classes of compounds before the name of the compound, for example, lines 293-295 "a diterpenoid: phytol (46), two triterpenoids: α-amyrin (47), and betulinic acid (48), and a flavonoids: acacetin (50), a sesquiterpenes: 12α,4α-dihydroxybishopsolicepolide (39), and a coumarin : scopoletin (61)." remove the highlighted words and all the text.

Response 21: We have removed all the highlights as Reviewer’s advices (line 285, page 20 and lines 287-290 page 20; line 311, page 21; line 385 page 22; line 448, page 24).

Point 22: line 613-authors added caution to using Artemisia species for humans. the authors may need to explain more, and if any of the species reported to be toxic to human

Response 22: No toxicity of the Artemisia species for humans has been reported. However, no comprehensive toxicity testing of the Artemisia species in animals and humans has been conducted. Thus, we mention this caution in the Conclusions because of a precaution (line 615-616, page 27).

Reviewer 4 Report

  1. The review provides valuable information on the potential of Artemisia plants for the development of new anti-bacterial compounds. The authors have conducted a comprehensive literature search of articles published in various databases over a period of 19 years. The review highlights the significant progress made in understanding the phytochemistry, pharmacology, and toxicology of Artemisia plants.
  2. The authors have identified 20 species of Artemisia and 75 compounds that possess anti-bacterial functions and have discussed their possible mechanisms of action. The discussion on the chemistry (structure and plant species source) of the phytochemicals and their anti-bacterial activities is informative. The authors have also provided an overview of new anti-bacterial strategies using plant compounds and extracts.
  3. However, it would be helpful if the authors could provide a more critical analysis of the literature reviewed and discuss the limitations and challenges faced in the development of new anti-bacterial compounds from Artemisia plants. Additionally, it would be useful if the authors could highlight any gaps in knowledge or areas for future research in this field.
  4. Overall, the review provides a useful summary of the current state of research on the anti-bacterial potential of Artemisia plants and their phytochemicals.

Minor editing of English language required

Author Response

Dear Reviewer,

We thank you for your precious time and efforts to help us improve our manuscript. First of all, we provided the point-to-point reply to the reviewers’ comments. Please note that all the modifications are indicated based on the lines and pages (or the serial number (SN) for the table) of the clean version of revised manuscript (Artemisia-2402571-clean). Besides, the tracked version of the revised manuscript (Artemisia-2402571-Track-change) and the PTP_reply is provided as additional files for your reference. Thanks again for your great help.

Reviewer 4

Point 1: The review provides valuable information on the potential of Artemisia plants for the development of new anti-bacterial compounds. The authors have conducted a comprehensive literature search of articles published in various databases over a period of 19 years. The review highlights the significant progress made in understanding the phytochemistry, pharmacology, and toxicology of Artemisia plants.

 Response 1: Thanks Reviewer.  

Point 2: The authors have identified 20 species of Artemisia and 75 compounds that possess anti-bacterial functions and have discussed their possible mechanisms of action. The discussion on the chemistry (structure and plant species source) of the phytochemicals and their anti-bacterial activities is informative. The authors have also provided an overview of new anti-bacterial strategies using plant compounds and extracts.

Response 2: Thanks Reviewer.

Point 3: However, it would be helpful if the authors could provide a more critical analysis of the literature reviewed and discuss the limitations and challenges faced in the development of new anti-bacterial compounds from Artemisia plants. Additionally, it would be useful if the authors could highlight any gaps in knowledge or areas for future research in this field

Response 3: We have discussed the limitations and challenges of Artemisia plants in drug development and future research below.

The limitations and challenges faced in the development of new anti-bacterial compounds from Artemisia plants include 1) Artemisia plant-derived phytocompounds generally have low anti-bacterial potency, 2) anti-bacterial mechanisms of action of the compounds are not clear, 3) anti-bacterial compounds with high potency need more labor and time to be identified, and 4) total synthesis of the anti-bacterial com-pounds like terpenoids is challenging [87,88] (lines 587-592, pages 27).

Additionally, Artemisia plants have been used to treat bacterial infections in humans from ancient time. So far, only artemisinin from A. annua has been developed as a pre-scription drug against malaria. Some gaps in development of anti-bacterial drugs from the Artemisia plants include the identification, safety, efficacy, and synthesis of active compounds from this genus. Currently, components of the essential oils form this ge-nus was successfully identified and they had a MIC of 0.1 μg/mL against certain bacte-ria but not the other. This suggests other potential anti-bacterial compounds need to be characterized, which may reflect the gaps from current findings to drug discovery [89,90] (lines 593-600, pages 27)

Point 4: Overall, the review provides a useful summary of the current state of research on the anti-bacterial potential of Artemisia plants and their phytochemicals

 Response 4: Thanks Reviewer.

Point 5: Minor editing of English language required

Response 5: This revised manuscript has been edited by an English native speaker certified by the Board of Editor's in the Life Sciences (BELS).